# Rad51 determines pathway usage in post-replication repair

**Damon Meyer** [1,5,13], **Steven K. Gore**[1,13], **Jie Liu** [1,13], **Shannon J. Ceballos**[1,6,13], **Shih-Hsun Hung**[1,7], **Giordano Reginato** [2,8], **Maria I. Cano-Linares** [3], **Katarzyna H. Maslowska** [4,9], **Florencia Villafañez**[4,10], **Clare Fasching**[1,11], **Christopher Ede**[1,12], **Vincent Pagès** [4], **Felix Prado** [3], **Petr Cejka** [2] & **Wolf-Dietrich Heyer** [1] ✉

Stalled replication forks are processed in post-replication repair by homologous recombination, fork regression, and translesion DNA synthesis. However, the regulation of pathway usage is not fully understood. Rad51 protein maintains genomic stability through its roles in recombination and in protecting stalled replication forks. We report isolation of mutations in *Saccharomyces cerevisiae* Rad51 that shift post-replication repair from recombination to alternate pathways including mutagenic translesion synthesis. Rad51-E135D and Rad51-K305N show near normal in vitro recombination despite changes in their DNA binding profiles, in particular to dsDNA. The mutants lead to a defect in Rad51 recruitment to stalled forks in vivo as well as a defect in the protection of dsDNA from degradation by Dna2-Sgs1 and Exo1 in vitro. Together, the evidence suggests that Rad51 binding to duplex DNA is critical to control pathway usage at stalled replication forks.

Homologous recombination (HR) provides a central pathway to maintain genomic stability[1]. In the context of double-strand break (DSB) repair in mitotically growing cells, HR uses the intact sister chromatid as a template for DNA repair synthesis[2]. Initially, the DSB is resected in a 5′−3′ direction to generate 3′-OH ending single-stranded DNA tails[3]. Resection is initiated by the Mre11 nuclease in the Mre11-Rad50-Xrs2 complex, which, in conjunction with Sae2, can also process chemically complex DSB including those covalently bound to proteins. After an initial endonucleolytic incision and 3′−5′ resection by the Mre11 nuclease, two long-range nuclease pathways operate in the 5′−3′ direction including the dsDNA-specific exonuclease Exo1 and the ssDNA nuclease Dna2, which functions in conjunction with the helicase-topoisomerase complex Sgs1-Top3-Rmi1 that generates the ssDNA substrate for Dna2. Resection involves 100 s if not 1000 s of nucleotides of ssDNA which is bound by the ssDNA-binding protein RPA. Nucleation of Rad51 filaments on RPA-coated ssDNA is achieved by mediator proteins, mainly Rad52 in budding yeast, a role that is usurped by the BRCA2 tumor suppressor protein in mammalian cells[4]. The resulting Rad51-ssDNA filament catalyzes the signature HR steps of homology search and DNA strand invasion to generate displacement loops (D-loops) and prime repair DNA synthesis in conjunction with the dsDNA motor protein Rad54[4]. In budding yeast, Rad54 is essential

[1]Department of Microbiology & Molecular Genetics, University of California, Davis, CA, USA. [2]Institute for Research in Biomedicine, Faculty of Biomedical Sciences, Università della Svizzera italiana (USI), Bellinzona, Switzerland. [3]Centro Andaluz de Biología Molecular y Medicina Regenerativa –CABIMER; Consejo Superior de Investigaciones Científicas; Universidad de Sevilla; Universidad Pablo de Olavide, Seville, Spain. [4]Cancer Research Center of Marseille: Team DNA Damage and Genome Instability, CNRS, Aix Marseille Univ, Inserm, Institut Paoli-Calmettes, Marseille, France. [5]Present address: California Northstate University, College of Health Sciences, Rancho Cordova, CA, USA. [6]Present address: Turtle Tree, Woodland, CA, USA. [7]Present address: Institute of Biochemical Sciences, National Taiwan University, Taipei, Taiwan, ROC. [8]Present address: Laboratory for Cell Biology and Genetics, Rockefeller University, New York, NY, USA. [9]Present address: Institute of Molecular Biology (IMB), Mainz, Germany. [10]Present address: OncoPrecision, Córdoba, Argentina. [11]Present address: Melio Labs, Santa Clara, CA, USA. [12]Present address: Fulgent Genetics, Temple City, CA, USA. [13]These authors contributed equally: Damon Meyer, Steven K. Gore, Jie Liu, Shannon J. Ceballos. ✉e-mail: wdheyer@ucdavis.edu

to D-loop formation in vivo and in vitro[5,6]. The HR pathway, including Rad51, Rad52, and Rad54, also supports DNA replication by recovering broken replication forks in a process called break-induced replication and through the repair of replication-associated single-stranded gaps that occur after repriming[7–10].

Rad51 is the eukaryotic homolog of bacterial RecA with highly similar functions in HR[4,11,12]. RecA, however, has at least two additional functions in bacteria. It acts as a cofactor during translesion synthesis (TLS) by DNA PolV and it triggers the SOS response, a bacterial DNA damage response signaling pathway initiated by RecA-bound ssDNA[13,14]. Currently, no data suggests that Rad51 plays a similar role in eukaryotic TLS. A function of Rad51 bound to ssDNA in adaption to DNA damage has been postulated, but the mechanism of this potential role in DNA damage signaling has not been defined[15]. Eukaryotic Rad51 proteins bind ssDNA and dsDNA with similar affinity and only a slight preference for ssDNA[16–19]. This is in strong contrast to bacterial RecA, which strongly prefers binding to ssDNA[20,21]. Binding to ssDNA by Rad51 is essential for HR, whereas Rad51 binding to duplex DNA inhibits HR, providing the rationale for proteins that mediate the assembly of Rad51 filaments on ssDNA and the dissociation of Rad51 from dsDNA[16,22–26]. Rad51 binding to chromosomal duplex DNA may also be limited by chromatin and accessibility. This poses the question, why did high-affinity dsDNA binding evolve in eukaryotic Rad51 and what is its function?

An independent role of some HR proteins in response to replication stress has been identified in vertebrate cells in protecting nascent DNA at replication forks from pathological degradation by MRE11, DNA2 or EXO1[27–29]. Blockage of the replicative helicase or DNA polymerases and regressed forks generate substrates for these nucleases in the form of ssDNA/dsDNA junctions in gaps. In addition, fork reversal generates a DSB at the spur of the chicken foot structure. The HR proteins required for fork protection are BRCA2 and RAD51 but not RAD54[27,29], suggesting that RAD51 filament formation but not the entire HR process is required for fork protection. This is consistent with separation-of-functions mutants in human RAD51 and BRCA2 that affect differentially the HR and fork protection functions[27,30]. The mechanisms involved and the exact binding sites for BRCA2 and RAD51 to protect stalled replication forks are not fully understood and recent work suggested an involvement of RAD51 binding to dsDNA[31,32]. It is unclear how fork protection relates to the response to replication stress in yeast.

Replication stress and fork stalling can be elicited by unusual DNA structures, protein-DNA complexes, limiting nucleotide pools, interference from transcription as well as endogenously and exogenously induced DNA damage[29]. Stalling at DNA damage sites on the leading and lagging DNA strands can be overcome by direct translesion synthesis (TLS) or downstream repriming leaving a post-replicative gap[9,29,33]. Such a gap can either be closed by TLS or a homologous recombination (HR)-based mechanism of template switching (TS) using the sister chromatid as a donor. These processes may be spatially and temporally uncoupled from the replication fork[34–38]. TLS skips the lesion through the engagement of specialized DNA polymerases[39]. In yeast, a key TLS polymerase is Pol zeta (Pol ζ), a four-subunit assembly of Pol31, Pol32, Rev7 and the catalytic subunit Rev3[40]. TLS is supported by monoubiquitylation of PCNA at lysine 164 by the Rad6 E2 ubiquitin conjugase in concert with the Rad18 E3 ubiquitin ligase[41,42]. HR is critical for TS and involves the formation of Rad51-ssDNA filaments that perform homology search and DNA strand exchange in conjunction with its co-factors the Rad51 paralogs Rad55-Rad57 and the dsDNA motor protein Rad54[9,10,43].

Unique to fork stalling at leading strand lesion is the process of fork regression, another mechanism of TS, where the fork regresses into a four-stranded structure called a chicken foot. Yeast Rad5 was shown to promote fork regression in vitro[44]. However, in vivo, DNA damage checkpoint signaling suppresses fork regression in yeast and

direct electron microscopic analysis detected few regressed forks in wild type cells after the addition of hydroxyurea (HU), an inhibitor of ribonucleotide reductase that leads to fork stalling[45]. In contrast, in mammalian cells, fork regression is frequently observed in response to various fork stalling agents including HU, methyl methanesulfonate (MMS) and UV, and this process is dependent on the central HR protein RAD51[46] as well as the human RAD5 homolog HLTF[47]. Contra-directional migration of the regressed fork can reestablish a replication fork with the original lesion being avoided. In human cells, fork regression and restart of regressed forks are controlled by a complex ensemble of dsDNA translocases (ZRANB3, SMARCAL1, HLTF) and DNA helicases (FBH1, RECQ1) regulated by poly-ubiquitylation of PCNA[29]. Alternatively, the chicken foot structure which creates a DSB could be processed by 5′–3′ exonucleases to generate a 3′-ending ssDNA strand available for Rad51 filament formation and DNA strand invasion in front of the fork stall point to restart DNA synthesis[48]. In sum, TS comprises several mechanistically diverse processes, including fork regression and several forms of gap repair, which can occur at the stalled fork or behind the stalled fork[49]. How the usage of these different pathways is managed is currently not fully understood. One aspect is PCNA poly-ubiquitylation by the ubiquitin-conjugating complex Ubc13/Mms2 and Rad5[49], but it is not clear whether all TS pathways are affected. Consistently, inactivation of PCNA poly-ubiquitylation by a *ubc13* mutation inhibits TS leading to a strong increase in the use of the TLS pathway[50]. Interestingly, Rad5 also provides an additional pathway to recruit TLS polymerases to stalled forks[51–55]. Another regulator in budding yeast appears to be the Srs2 helicase, which has the ability to strip Rad51 from ssDNA, an activity that is counteracted by Rad55-Rad57[56–58]. Interestingly, Srs2 is recruited to stalled forks by sumoylation of PCNA on lysines 127 and 164 thanks to its C-terminal SUMO binding motif[59,60]. The genetic observations that sensitivity to fork stalling agents caused by mutations in Rad5, Rad6 and Rad18 that abolish TLS and TS are largely suppressed by mutations in Srs2 are consistent with a model that Srs2 suppresses activation of an HR-dependent TS pathway independent of PCNA ubiquitylation (also termed salvage pathway) when recruited by SUMO-PCNA[61–63]. In sum, HR-dependent and independent TS as well as TLS pathways operate at regressed forks in yeast, but it remains unclear what regulates their usage[64,65].

To define what controls pathway usage at stalled replication forks in the budding yeast *Saccharomyces cerevisiae*, we isolated mutants in *RAD51*, which severely affect pathway usage at replication forks stalled by the alkylating agent MMS. The mutants display sensitivity to MMS, and their survival strongly depends on other post-replication repair pathways, namely TLS and fork regression. We directly show that pathway usage at fork stalling lesions is dramatically changed. The Rad51 mutant proteins are poorly recruited to stalled forks in vivo and cannot protect dsDNA from exonucleolytic attack in vitro. Biochemical analysis identified a strong defect in dsDNA binding as the major characteristic that correlated between the in vitro and in vivo phenotypes of the mutants. We suggest that Rad51 binding to dsDNA is a major determinant in post-replication repair and that the properties of the Rad51 filament are critical for pathway usage at stalled forks. While the mutants were proficient in spontaneous and DSB-induced sister chromatid recombination, they displayed reduced interhomolog recombination likely caused by a defect in forming extended filaments on DNA, suggesting different requirements for inter-sister recombination versus a genome-wide homology search.

## Results
### *RAD51* mutant isolation
Rad51 protein functions as the central homology search and DNA strand invasion factor during HR, which involves filament formation on ssDNA. In an independent role, Rad51 protects stalled replication forks, but the mechanisms involved are less well defined. Rad54 is a dsDNA

## A  Mutant isolation

## B  Reconstructed validation

**Fig. 1 | Isolation, validation, and MMS sensitivity of *rad51-ED* and *rad51-KN*.**
**A** Scheme to isolate *rad51* mutants with predicted phenotypes for wild type cells (WT), *rad51Δ rad54ts* cells with wild type *RAD51* (*RAD51*) and a candidate mutant (candidate). **B** Validation of mutant phenotype with plasmid-borne gene by transformation into fresh tester strains. Serial dilution *assays* of *rad51* mutations at 37 °C in the presence and absence of 0.002% MMS (*rad51Δ rad54ts*: WDHY2546 with pWDH957 (*RAD51*), pWDH954 (*rad51-ED*), pWDH953 (*rad51-KN*); *rad51Δ rad54Δ*: WDHY2544 with pWDH957 (*RAD51*), pWDH954 (*rad51-ED*), pWDH953 (*rad51-KN*)). **C−E** MMS sensitivity of chromosomally integrated mutations. Survival of cells

following exposure to varying levels of MMS (0.0025%, 0.005%, 0.0075%, 0.01%, 0.02%, and 0.03%) was measured as the number of colonies formed on YPD relative to unexposed cells (0% MMS) in the following genotypes: wild type (WT) (WDHY3960), *rad51Δ* (WDHY3898), *rad54Δ* (WDHY3961), *rad51Δ rad54Δ* (WDHY3959), *rad51-KN* (WDHY3962), *rad51-ED* (WDHY3548), *rad51-KN rad54Δ* (WDHY3963), and *rad51-ED rad54Δ* (WDHY3547). The mean and SEM were calculated from a minimum of three independent cultures. Source data are provided as a Source Data file.

motor protein that is critical for HR but not required for fork protection in mammalian cells[27]. Using sensitivity to the fork stalling agent MMS, we isolated two mutations in *RAD51*, each with a single amino acid change, *rad51-E135D* (referred to as *rad51-ED*) and *rad51-K305N* (referred to as *rad51-KN*), that suppress the MMS-sensitivity of *rad51 rad54* double mutants (Fig. 1A). In high copy when borne on a 2 μ plasmid, both *rad51* mutants substantially suppressed the MMS sensitivity of a *rad51Δ rad54ts* double mutant at the restrictive temperature. There was also significant but less extensive suppression of the MMS sensitivity of a *rad51Δ rad54Δ* double mutant (Fig. 1B). We surmise that the *RAD51* mutants change Rad51 filament properties that allows use of alternate post-replication repair pathways at stalled forks to bypass the block in HR imparted by the *RAD54* mutation.

### *rad51-ED* and *rad51-KN* cause sensitivity to fork stalling
To study these mutations in a more controlled fashion, we introduced them into the chromosomal *RAD51* gene and conducted survival assays through colony formation (Fig. 1C–E). We found the expected high sensitivity of *rad51Δ* and *rad54Δ* mutant cells to MMS and the expected epistasis between both mutations (Fig. 1C). The *rad51-ED* mutant was as sensitive to MMS as the *rad51Δ* strain, whereas the *rad51-KN* was only mildly more sensitive than wild type cells, notable only at higher MMS doses (Fig. 1D). Neither chromosomal mutant was able to suppress the MMS sensitivity of a *rad54Δ* mutation in the colony formation assay (Fig. 1E), suggesting that the overexpression during mutant isolation was a factor. We also note that the colony formation assay (Fig. 1C–E) measures survival whereas the serial dilution assay used in the screen and validation (Fig. 1B) measures a sum of growth and survival. The steady-state levels of the Rad51-ED and Rad51-KN mutant proteins were somewhat elevated compared to wild type Rad51 protein (Supplementary Fig. 1), eliminating the possibility that low protein levels are the cause of the observed phenotypes. We conclude that Rad51-ED and Rad51-KN are mutant proteins with altered functions in the cellular response to MMS.

### *rad51-KN* is proficient in sister chromatid recombination while *rad51-ED* has a subtle defect
To assess the recombination properties of the newly isolated *rad51* mutants, we employed a well-established sister chromatid recombination assay that allows monitoring of spontaneous and DSB-induced recombination between two directly repeated copies of mutant *leu2* genes that can give rise to Leu+ recombinants (Fig. 2A). The system uses *URA3* as a marker between the repeated *leu2* genes to distinguish gene conversion events (Leu+ Ura+) from other events that include single-strand annealing, intra-chromatid crossing over, unequal sister-chromatid exchange, unequal sister-chromatid conversion, and replication slippage and are summarily called pop-out events (Leu+, Ura−). Spontaneous pop-out events in this system are largely mediated by Rad52-dependent strand annealing and are independent of Rad51 (ref. 66 and Fig. 2B left). Neither *rad51Δ* and *rad54Δ* nor *rad51-ED* and *rad51-KN* mutations affected these events. The Leu+ Ura+ gene conversion events strongly depended on Rad51, Rad52, and Rad54, as expected (Fig. 2B right). The rate for spontaneous Leu+ Ura+ events in *rad51-ED* was slightly but statistically significantly reduced (p = 0.017), while *rad51-KN* (p = 0.084) was not. The mutants did not suppress the recombination defect caused by the *rad54Δ* mutation (Fig. 2B right). The frequency of both events (Leu+ Ura+, Leu+ Ura−) can be dramatically stimulated by the induction of a DNA double-stranded break using a galactose-controlled version of the HO-endonuclease which cleaves a HO recognition site engineered between the repeated mutant *leu2* genes (Fig. 2A). In determining the frequencies of DSB-induced pop-out (Leu+ Ura−) and gene conversion (Leu+ Ura+) events, the results showed that the *rad51-ED* and *rad51-KN* mutants exhibited no interference with pop-out events, no defect in DSB-induced gene conversion, and no suppression of the *rad54Δ* gene conversion defect (Fig. 2C). We conclude that *rad51-KN* is proficient for mitotic sister chromatid recombination, while *rad51-ED* has a subtle defect in spontaneous gene conversion but is fully proficient for DSB-induced events.

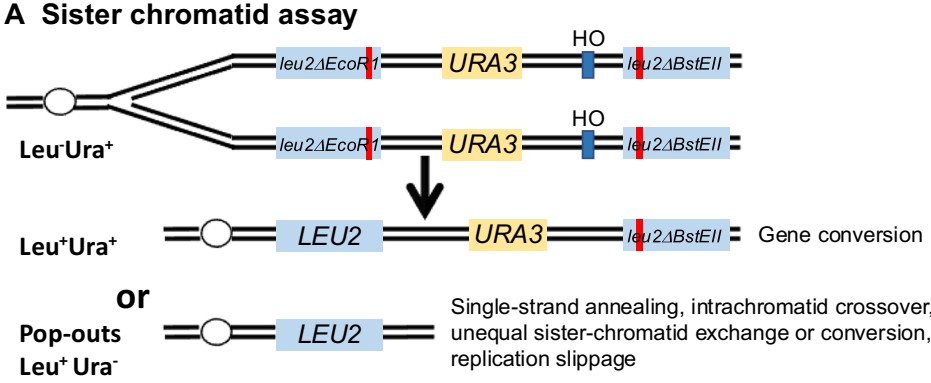

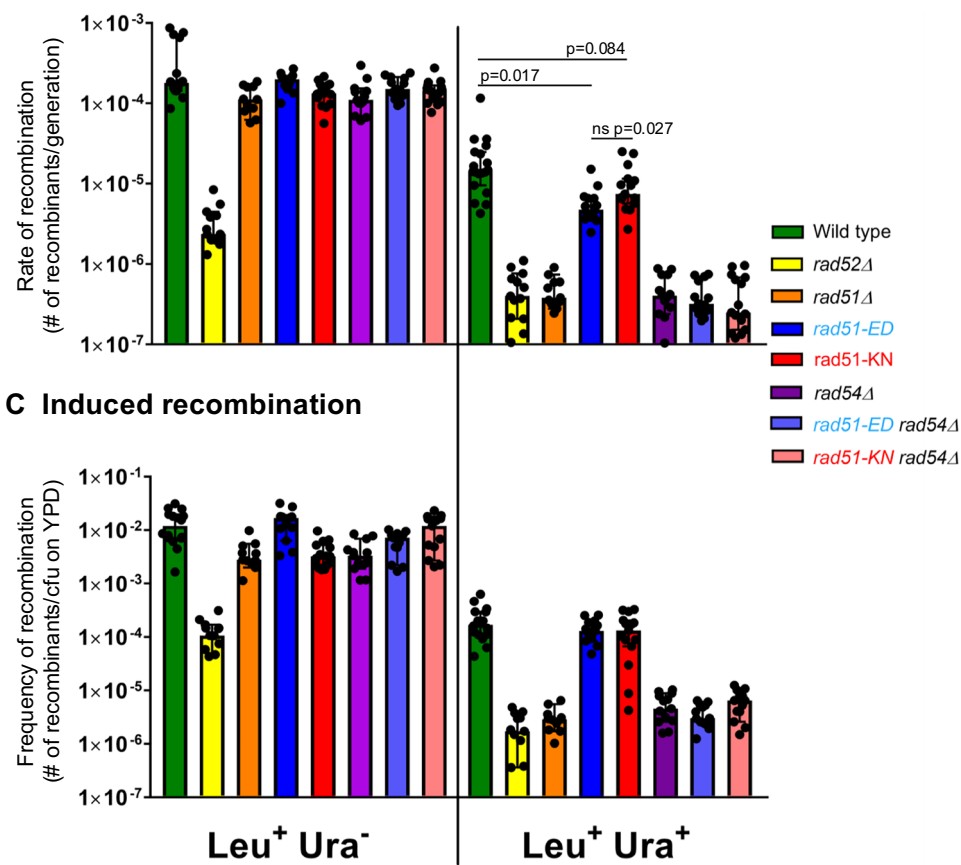

**Fig. 2 | *rad51-ED* and *rad51-KN* are recombination-proficient and cannot suppress the recombination defect of *rad54*. A** Direct repeat recombination assay[66]. HO: HO endonuclease cleavage site. **B** Spontaneous recombination rate data. **C** Recombination frequency data after DSB induction. Strains were freshly dissected from diploid strains and single spore clones were assayed, a minimum of 11–16 single clones were assayed per genotype, specific values for n are given with the genotypes. The median recombination rates/frequencies are given with the 95% confidence intervals[100]. The indicated p-value is derive from a two-tailed t-test.

Wildtype (WDHY3383, *n* = 16 in B, *n* = 16 in C), *rad52Δ* (WDHY3348, *n* = 13, 13), *rad51Δ* (WDHY3915, *n* = 11, 11)*, rad51-KN* (WDHY3385, *n* = 15, 15), *rad51-KN rad54Δ* (WDHY3463, *n* = 15, 13), *rad51-ED* (WDHY3386, *n* = 15, 14), *rad51-ED rad54Δ* (WDHY3462, *n* = 13, 13)*, rad54Δ* (WDHY3349, *n* = 13, 13). The viability data are shown in Supplementary Fig. 2. The data are all relative to the number of colony forming units (cfu) on YPD. Relevant p values are given. ns: not significant. No other differences involving *rad51-ED* and *rad51-KN* were significant. Source data are provided as a Source Data file.

While *rad52Δ* showed a significant survival defect after DSB induction by HO, *rad51Δ* and *rad54Δ* showed a smaller survival defect (Supplementary Fig. 2), as expected. This reflects the involvement of Rad52 in both pop-out and gene conversion events, whereas pop-out events are unaffected in *rad51Δ* and *rad54Δ* cells, allowing substantial DSB repair. The *rad51-ED* mutant shows slightly better survival after DSB induction than *rad51Δ* cells which was within the error of the experiment, whereas *rad51-KN* cells survived significantly better than

*rad51Δ* (Supplementary Fig. 2). This difference in response to a single DSB between both mutants was also recapitulated by their differential sensitivity to 100 Gy of ionizing radiation in serial dilution plate assays, where *rad51-ED* was IR sensitive but somewhat less so than *rad51Δ* (Supplementary Fig. 3A, compare rows 1 and 5). *rad51-KN* showed resistance to IR that was equivalent to wild type cells (Supplementary Fig. 3A, compare rows 5 and 9). We conclude that *rad51-ED* has a survival defect in response to a single DSB or IR despite being fully

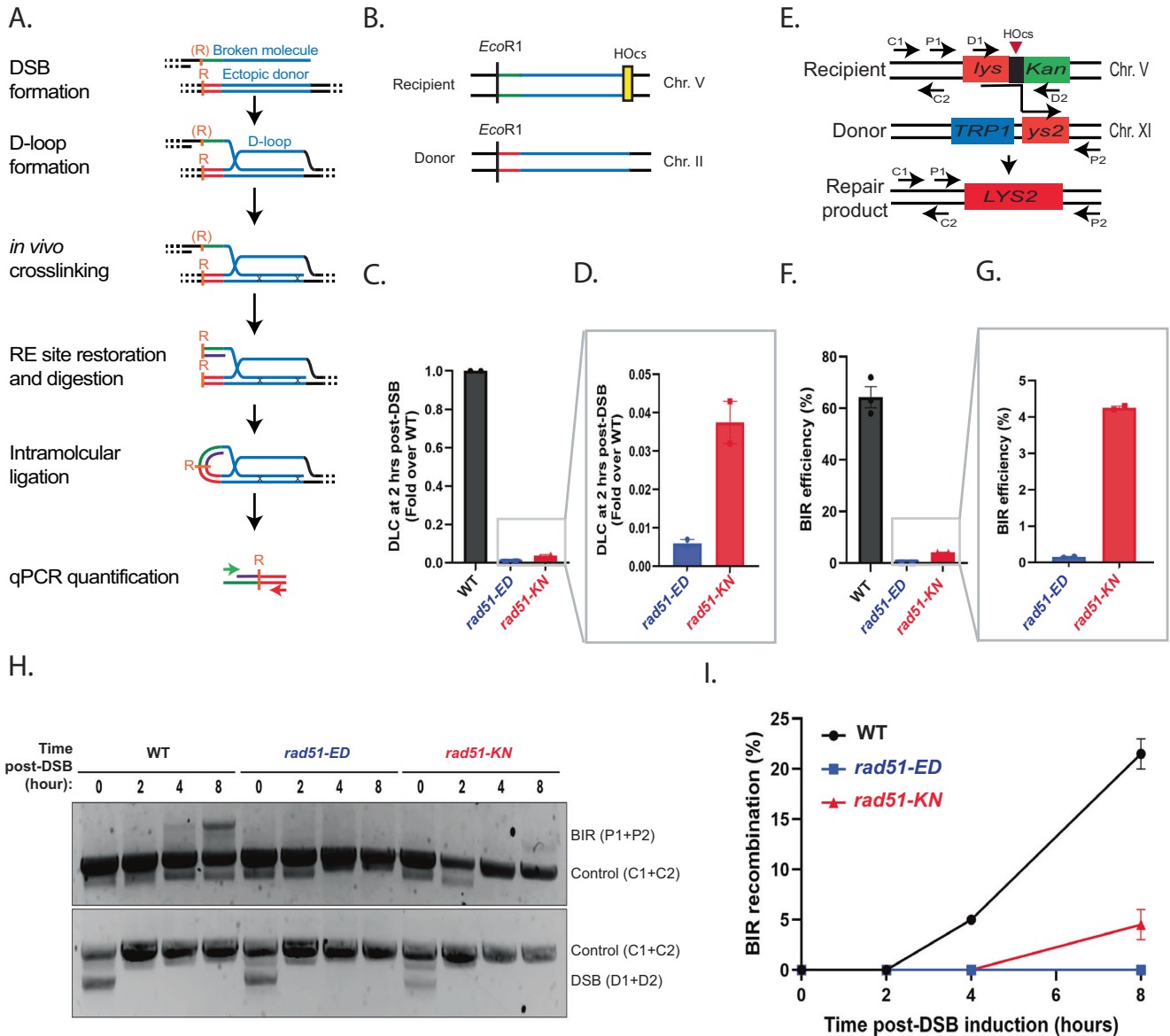

**Fig. 3 | rad51-ED and rad51-KN are defective in inter-homolog recombination.**
**A** Schematic representation illustrating the D-loop capture (DLC) assay.
**B** Schematic representation depicting the construct of the DLC assay strain.
**C** Quantification of the DLC signal from the wild type (WDHY5509), *rad51-ED*
(WDHY6283) and *rad51-KN* (WDHY6288) strains obtained at 2 h HO induction.
Error bars, SEM (*n* = 2). **D** Zoomed-in comparison of the DLC signal between *rad51-ED* (WDHY6283) and *rad51-KN* (WDHY6288) strains obtained at 2 h post HO
induction. Error bars, SEM (*n* = 2). **E** Schematic representation illustrating the
construct for the Break-Induced Replication (BIR) assay. **F** Quantification of the BIR
efficiency from the wild type (WDHY6298), *rad51-ED* (WDHY6304) and *rad51-KN*
(WDHY6299) strains. Error bars, SEM (*n* = 3 for wild type, *n* = 2 for *rad51-ED*, *rad51-*

*KN*), ***: P < 0.0004. **G** Zoomed-in comparison of BIR efficiency between *rad51-ED*
(WDHY6304) and *rad51-KN* (WDHY6299). Error bars, SEM (*n* = 2). **H** Agarose gel
analysis was conducted to assess PCR products from various primer sets, exam-
ining the kinetics of Break-Induced Replication (BIR) product formation (P1 and P2),
DNA loading control (C1 and C2), and DSB cleavage efficiency (D1 and D2) from the
indicated strains at various time points following HO induction. **I** Plot illustrating
the quantification of Break-Induced Replication (BIR) products at each time point,
as obtained from (**H**). Statistical analysis of DLC and BIR results for each mutant was
compared with their respective paired mutants using a two-way ANOVA in Prism 10
(GraphPad Software). Source data are provided as a Source Data file.

proficient in DSB-induced inter-sister HR. This may be related to the
signaling role of DNA bound Rad51 affecting adaptation from DNA
damage[15]. IR also induces base damages that may elicit replication fork
stalling leading to a possible alternate interpretation that the IR-
sensitivity of the mutants is related to fork protection (see below),
although we did not further explore the IR phenotype.

### *rad51-ED* and *rad51-KN* exhibit a defect in interhomolog recombination

To further define the HR phenotypes of the *rad51-ED* and *rad51-KN*
mutants we conducted a series of physical and genetic HR assays that

require genome-wide homology search. To directly test the ability of
the mutant protein to form D-loops in vivo, we first employed the
D-loop capture (DLC) assay[6]. In short, this assay physically monitors
D-loop formation between an HO-induced DSB on chromosome V with
a homologous donor on chromosome II (Fig. 3A, B) by capturing the
D-loop through psoralene-crosslinking and using proximity ligation
between two *EcoRI* sites adjacent to the DSB and donor site to quantify
by qPCR the amounts of D-loops. The DLC signal completely depends
on Rad51, with an over 100-fold reduction in signal in the *rad51Δ*
mutant[6]. The *rad51-ED* and *rad51-KN* displayed a strong defect in
D-loop formation, with *rad51-ED* being significantly more affected than

*rad51-KN* (Fig. 3C, D). We confirmed this observation in an independent assay measuring BIR between a DSB on chromosome V and a donor site on chromosome XI, which also showed a strong defect in both mutants with *rad51-ED* being more affected than *rad51-KN* in BIR efficiency measured through the genetic endpoint (LYS + ) and the physical product (Fig. 3E–I). We conclude that *rad51-ED* and *rad51-KN* lead to substantial defect in recombination requiring genome-wide homology search, despite their substantial proficiency in sister chromatid recombination.

## Rad51-ED and Rad51-KN proteins exhibit a strongly reduced Kd for dsDNA

Rad51-E135 is an invariable residue in a highly conserved region of the eukaryotic Rad51 proteins which is not conserved in the bacterial RecA homologs. Rad51-K305 is conserved in most Rad51 proteins in its charge (K or R), although some Rad51 proteins carry hydrophobic amino acids (L, A) at this position (Supplementary Fig. 5A). Both residues are not directly involved in DNA or nucleotide co-factor binding in the Rad51-ssDNA-ADP-AlF$_3$ structure[67] (Supplementary Fig. 5B), although K305 is part of an alpha helix that ends in R293 which intercalates between bases to enforce the triplet spacing of DNA bound to Rad51 (Supplementary Fig. 5C). To better understand the underlying biochemical defects and mechanisms involved, we purified the native mutant proteins to apparent homogeneity for biochemical analysis (Supplementary Fig. 5D). The mutant proteins purified like the wild type protein and were free of contaminating activities (endo/exonuclease, topoisomerase, phosphatase) that would interfere with the in vitro experiments and their interpretation.

The key biochemical activities of Rad51 are binding and filament formation on DNA, DNA-stimulated ATP hydrolysis, homology search and DNA strand invasion, which underlie its functions in HR and at stalled forks. First, we measured DNA binding of 100 nt and 100 bp substrates using electrophoretic mobility shift assays (EMSA) (Fig. 4). The binding to ssDNA by wild type Rad51 showed a Kd of 497 ± 39 nM for 50 nM NaCl and 578 ± 27 nM for 100 mM NaCl consistent with previous measurements[16]. Rad51-ED showed identical behavior (Kd=486 ± 12 nM for 50 mM NaCl, 602 ± 11 nM for 100 mM NaCl), whereas Rad51-KN had slightly higher affinity than wild type Rad51 with a Kd of 339 ± 10 nM for 50 mM NaCl and 402 ± 3.3 nM for 100 mM NaCl (Fig. 4A). The mutant complexes showed more of a smear, which is likely related to subtly reduced stability of the protein:ssDNA complexes. Importantly, in contrast to ssDNA, both mutants showed a very significant defect in binding to dsDNA, such that a Kd could not be calculated (Fig. 4B). The defects of both proteins were stronger at elevated salt concentration (100 vs. 50 mM NaCl). The Rad51-ED mutant protein showed consistently a stronger defect than the RAD51-KN protein. We extended these results using longer DNA substrate (ssDNA: ΦX174 viral DNA 5386 nt; dsDNA: ΦX174 RF1 5386 bp) by conducting salt midpoint titrations (Supplementary Fig. S6). The salt titration curves allowed the calculation of salt titration midpoints (STM; Supplementary Fig. 6E) as the salt concentration, at which 50% of the protein-DNA complexes are formed. A lower STM signals lower affinity for DNA. Under no salt conditions (0 mM NaCl), both mutant proteins showed no defect in binding to ssDNA and a small but significant defect for dsDNA (Supplementary Fig. 6C, D). The defect for Rad51-ED was again significantly stronger than with Rad51-KN for both substrates. In fact, at physiological ionic strength (-120 mM NaCl[68]), dsDNA binding was largely suppressed, especially for Rad51-ED. In contrast to the 100 nt substrate, the mutants showed a defect in ssDNA binding with the longer substrate, which is explained by the formation of some duplex regions in the more complex ssDNA substrate and possibly by reduced stability of ssDNA binding.

To further explore the DNA binding defect, we examined the Rad51 filaments on 1000 bp dsDNA and 700 nt ssDNA substrates by electron microscopy (EM). Unexpectedly, we saw a significant defect

of both mutants to form extended nucleoprotein filaments on both substrates (Supplementary Fig. 7). As in the EMSA and STM assays, the defect in dsDNA binding was more pronounced than with ssDNA (compare Supplementary Fig. 7D with H). The defect with rad51-KN was more pronounced than with Rad51-ED. This defect correlated with severely diminished Rad51 focus formation in response to MMS as observed by immunofluorescence (Supplementary Fig. 8). It is unclear whether the MMS-induced Rad51 foci represent Rad51 bound to ssDNA, dsDNA or both, but the results suggest that the filament length of both mutants in cells is below the detection threshold in our experiment.

## Rad51-ED and Rad51-KN proteins have altered ATPase profiles

The ATPase of Rad51 is strongly dependent on DNA binding, and an ATPase defect could be a consequence of an underlying DNA binding defect. We measured the ATPase activity of the mutant proteins along with wild type Rad51 at 200 mM NaCl to accentuate potential differences. The ATPase activities of the mutant proteins were diminished with both ssDNA and dsDNA co-factors. The defect of the Rad51-ED mutant protein was significantly more pronounced than that of the Rad51-KN mutant (Fig. 5A, B).

## Rad51-ED and Rad51-KN proteins are proficient in D-loop formation

The in vitro recombination activity of the mutant proteins was assessed in the classic D-loop assay using a tailed 100mer with a 5' 25 bp duplex region and a 2686 bp negatively supercoiled duplex target DNA in reconstituted reactions with Rad51, Rad54 and RPA conducted as a time course at the same salt concentrations (50 mM Supplementary Fig. 9A–C; 100 mM NaCl Fig. 5C–E) as the EMSA assays of Fig. 4. Rad51 wild type protein showed the expected behavior of forming D-loops peaking at 5 min that were processed over time (Fig. 5E). The mutant Rad51-ED and Rad51-KN proteins were as effective as wild type Rad51 in forming D-loop in vitro and the D-loops formed showed somewhat higher D-loop instability over time (Fig. 5E). This is consistent with a functional defect in binding dsDNA, as after D-loop formation Rad51 is bound to dsDNA, from where it is displaced for D-reversal by Rad54[69]. D-loop formation by the Rad51-ED and Rad51-KN proteins was fully dependent on Rad54, identical to wild type Rad51 (Supplementary Fig. 9D), as expected from the genetic experiments (Fig. 2). We conclude that the Rad51-ED and Rad51-KN mutant proteins are proficient in D-loop formation, consistent with the in vivo inter-sister recombination data (Fig. 2).

In sum, the biochemical analysis of the Rad51-ED and Rad51-KN proteins showed that they are proficient in forming D-loops in vitro and largely proficient in binding ssDNA. The major defect is in binding to dsDNA, which we further explored below.

## Rad51-ED and Rad51-KN fail to protect dsDNA from Exo 1 or Dna2

Besides its key function in HR by forming the presynaptic filament on ssDNA for homology search and DNA strand invasion, Rad51 protects nascent DNA at stalled DNA replication forks from unscheduled degradation by nucleases including Exo1 and Dna2. The fork protection function of certain HR proteins was first noticed in human cells, particularly in BRCA1/2-deficient backgrounds[27–29]. The DNA protection function of RAD51 is distinct from its role in HR, as indicated by the absence of a requirement for RAD54[27]. However, the fork protection function of RAD51 involves its binding to dsDNA[31,32].

To define the capacity of the yeast Rad51 variants in DNA protection, we employed in vitro assays with 100 bp-long dsDNA (Fig. 6A). We first used yeast Exo1, one of the nucleases involved in long-range DNA end resection. Without Rad51, Exo1 degraded -90% of the DNA substrate. Wild type Rad51 inhibited DNA degradation by Exo1, while the two Rad51 mutants, Rad51-KN and Rad51-ED, did not protect DNA

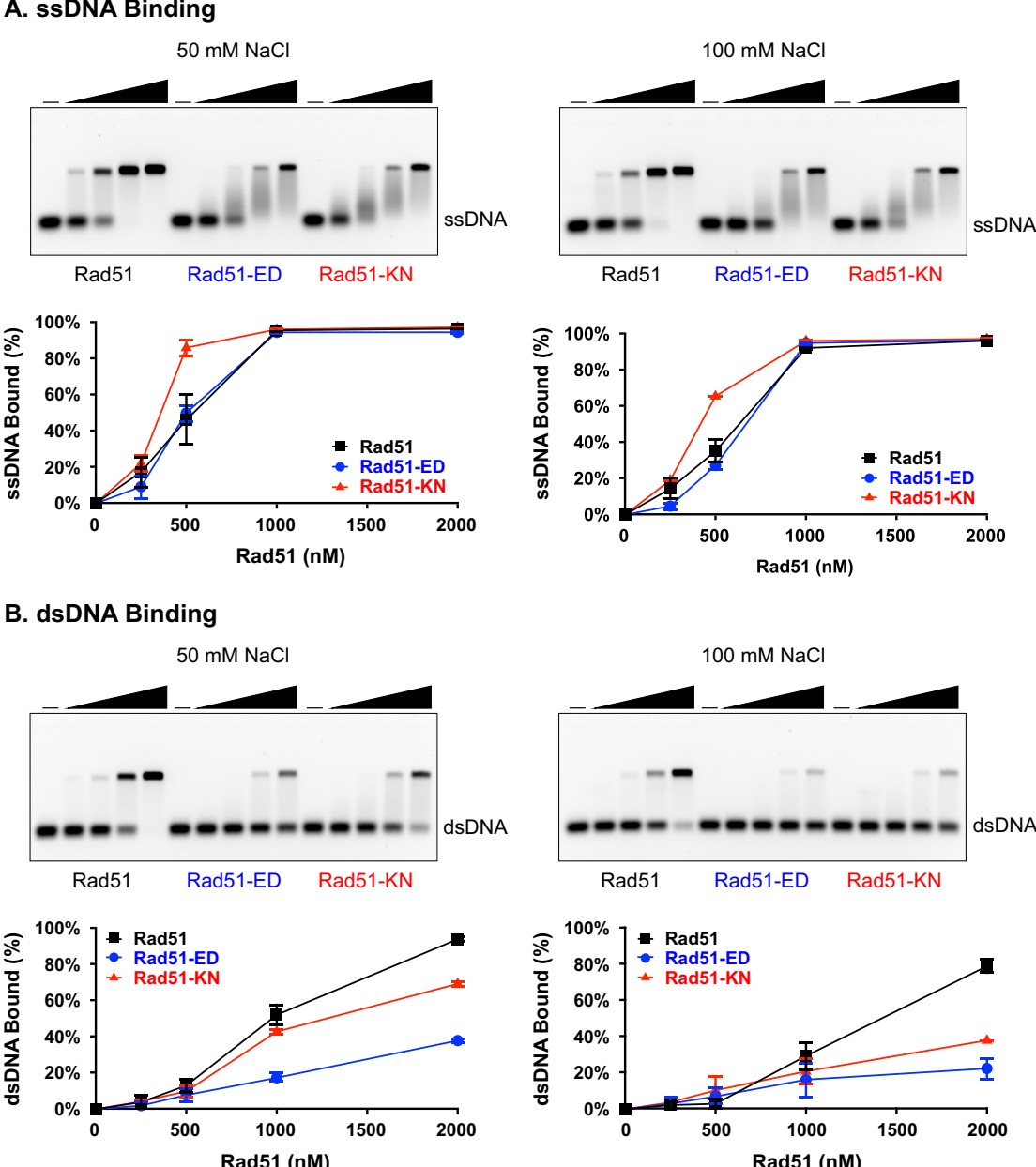

**Fig. 4 | Rad51-ED and Rad51-KN mutant proteins show salt sensitive reduced dsDNA binding.** Rad51 DNA binding was determined with wild type Rad51 (black rectangles), Rad51-ED (blue circles) and Rad51-KN (red triangles) in the presence of ssDNA (3 μM 100 nt ssDNA in **A**) or dsDNA (3 μM 100 bp dsDNA in **B**). Shown are means of 3 independent experiments ($n = 3$), error bars represent the standard deviation but are not always visible as they were smaller than the plotting symbol. Source data are provided as a Source Data file.

(Fig. 6B, C). Similar results were obtained when we used the Sgs1-Dna2 helicase-nuclease pair. While wild type Rad51 efficiently prevented DNA degradation, Rad51-KN and Rad51-ED were much less effective (Fig. 6C, D). Mechanistically, Rad51 prevents DNA unwinding by Sgs1, which prevents the generation of ssDNA for the Dna2 nuclease (Supplementary Fig. 10). Rad51-KN and Rad51-ED did not prevent unwinding (Supplementary Fig. 10B, C). Together, our reconstitution experiments revealed that Rad51-KN and Rad51-ED are impaired in dsDNA protection.

### Synergy with defect in post-replication repair

The biochemical analysis revealed a significant dsDNA binding defect in the mutants and a defect in protecting dsDNA from nuclease degradation. These properties are associated with replication fork stabilization by Rad51 and prompted us to explore the genetic interactions of *rad51-ED* and *rad51-KN* with mutations affecting various aspects of post-replication repair. We first tested a deletion mutation of *REV3*, which by itself shows no sensitivity to MMS at the low levels (0.0033%) used for the serial dilution assays in this experiment (Fig. 7A row 3). This contrasts with the significant sensitivity of HR-defective mutants (*rad54Δ* row 2, *rad51Δ* row 5, *rad51Δ rad54Δ* row 6). An HR defect displayed strong synergy with a defect in *REV3* leading to extreme MMS sensitivity and also affecting growth/survival on YPD growth media without MMS (*rad54Δ rev3Δ* row 4, *rad51Δ rev3Δ* row 7, *rad51Δ rad54Δ rev3Δ* row 8). These results were expected and show that HR and TLS represent parallel competing pathways to process MMS-induced DNA damage, and the single mutant phenotype suggests that HR plays a more prominent role in wild type cells than TLS.

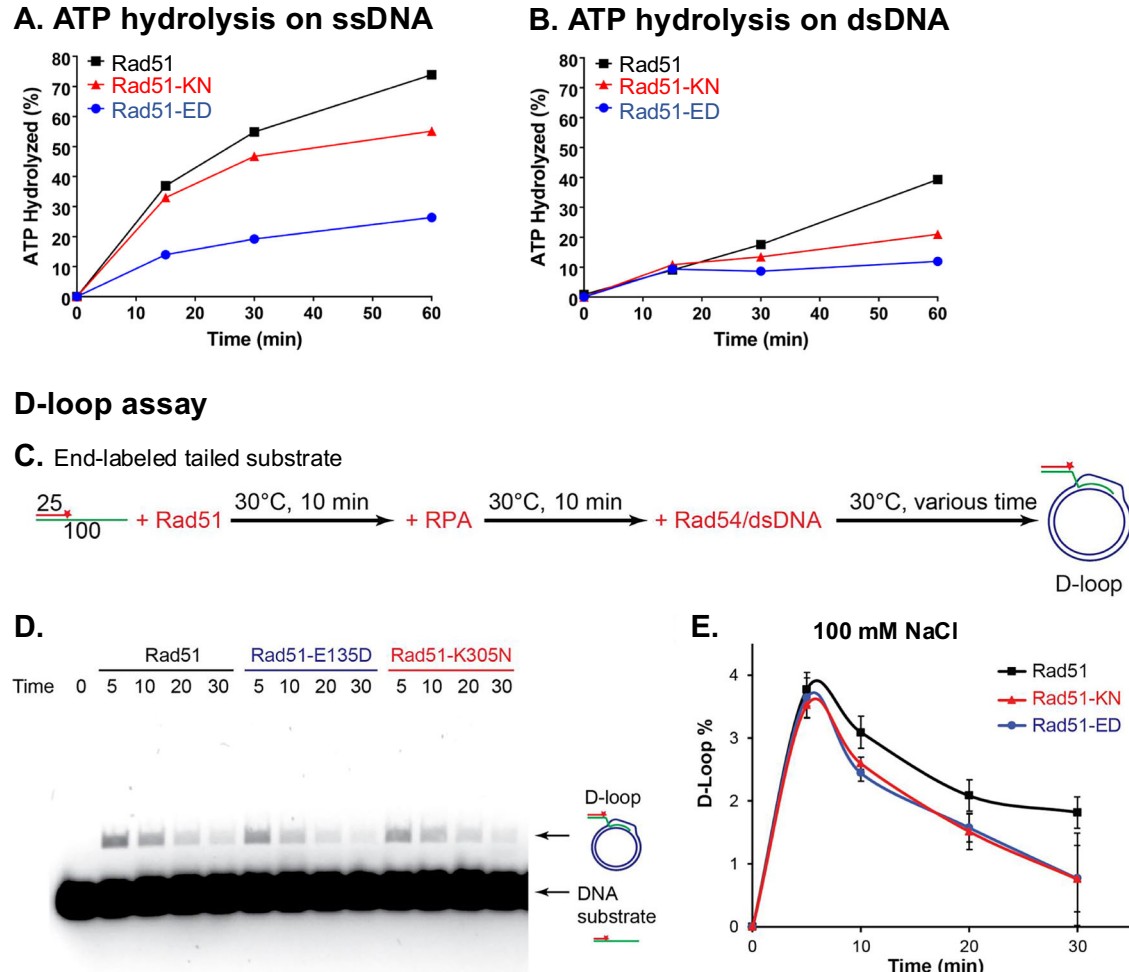

**Fig. 5 | Rad51-ED and Rad51-KN mutant proteins show reduced ATPase activity but are proficient in D-loop formation.** ATPase rates were determined using 3 μM of the purified proteins wild type Rad51 (black rectangles), Rad51-ED (blue circles) and Rad51-KN (red triangles) in the presence of ssDNA (9 μM ΦX174 virion DNA in **A**) or dsDNA (9 μM ΦX174 RF1 in **B**). Shown are means of n = 3, error bars represent the standard error but are not visible as they were smaller than the plotting symbol. **C** Schematic of the D-loop assay setup. A fluorescent tailed substrate was incubated with Rad51, Rad51-ED, and Rad51-KN, then mixed with RPA and Rad54+dsDNA, following the illustrated procedure. **D** Gel image showing D-loop formed by Rad51 variants in a physiologically relevant buffer containing 100 mM NaCl. **E** D-loop yield quantified from **B**, with means ± 1 sd from 3 independent experiments (*n* = 3). Source data are provided as a Source Data file.

*rad51-ED* showed the MMS sensitivity expected from the previous experiments (Fig. 1D, Supplementary Fig. 3A) that rivals that of a complete HR defect (Fig. 7A compare *rad54Δ* row 2, *rad51Δ* row 5, *rad51Δ rad54Δ* row 6 with *rad51-ED* row 9). The *rad51-ED rev3Δ* double mutant showed extreme MMS sensitivity and a growth/survival defect on YPD, like a complete HR defect (Fig. 7A compare *rad51-ED* row 9 with *rad51-ED rev3Δ* row 11). The addition of a *rad54Δ* mutation to these strains (rows 10, 12) did not further increase MMS sensitivity or the growth/survival defect on YPD. The low MMS concentration (0.0033%) was needed to compare the widely varying sensitivity of all strains on a single plate. *rad51-KN* showed little to no MMS sensitivity in the serial dilution assays (Fig. 7B row 9), as expected from Fig. 1, where this mutant showed MMS sensitivity only at high MMS concentrations. Remarkably, *rad51-KN* showed synergy with *rev3Δ* (Fig. 7 row 11), although none of the single mutants were measurably sensitive at the concentration used (rows 3, 9). These results show that MMS-stalled forks in *rad51-ED* cells are preferentially processed by TLS similar to the situation in cells with a complete HR defect, although this mutant is capable of normal HR. The *rad51-KN* mutant shows similar but less pronounced phenotypes.

Next, we tested the role of Mms2, which is required for poly-ubiquitylation of PCNA at K164[41] and template switching. In

quantitative survival assays, we confirmed the relative mild MMS survival defect of *mms2Δ* cells compared to HR-defective cells (*rad51Δ*, *rad54Δ*). Analysis of the *rad51Δ mms2Δ* and *rad54Δ mms2Δ* double mutants revealed a non-epistatic relationship between an HR defect and a loss of Mms2 function (Fig. 7C). These results show that in wild type cells HR makes a stronger contribution to MMS survival than pathway(s) dependent on PCNA poly-ubiquitylation. Moreover, these results suggest that Mms2 controls a strand exchange-independent template switching mechanism, in analogy to mammalian cells[70]. This mechanism is unlikely to be HR-mediated gap repair that is Rad51 dependent; alternatively, it could be fork regression operating in yeast in a Rad51-independent manner. *rad51-ED* showed the MMS sensitivity expected from the previous experiments (Fig. 1D) that rivals that of a complete HR defect. *rad51-ED* showed the same increase in MMS sensitivity in the double mutant with *mms2Δ* as the *rad51Δ* strain that remained unchanged by adding a *rad54Δ* mutation (Fig. 7D). *rad51-KN* showed little MMS sensitivity as expected from Fig. 1, which was similar to the MMS sensitivity of the *mms2Δ* single mutant (Fig. 7E). Remarkably, *rad51-KN* showed strong synergy with *mms2Δ* that was further enhanced by fully eliminating HR through a *rad54Δ* mutation (Fig. 7F). We conclude that MMS-stalled forks in *rad51-ED* cells have a higher propensity to be processed by a Mms2-dependent pathway

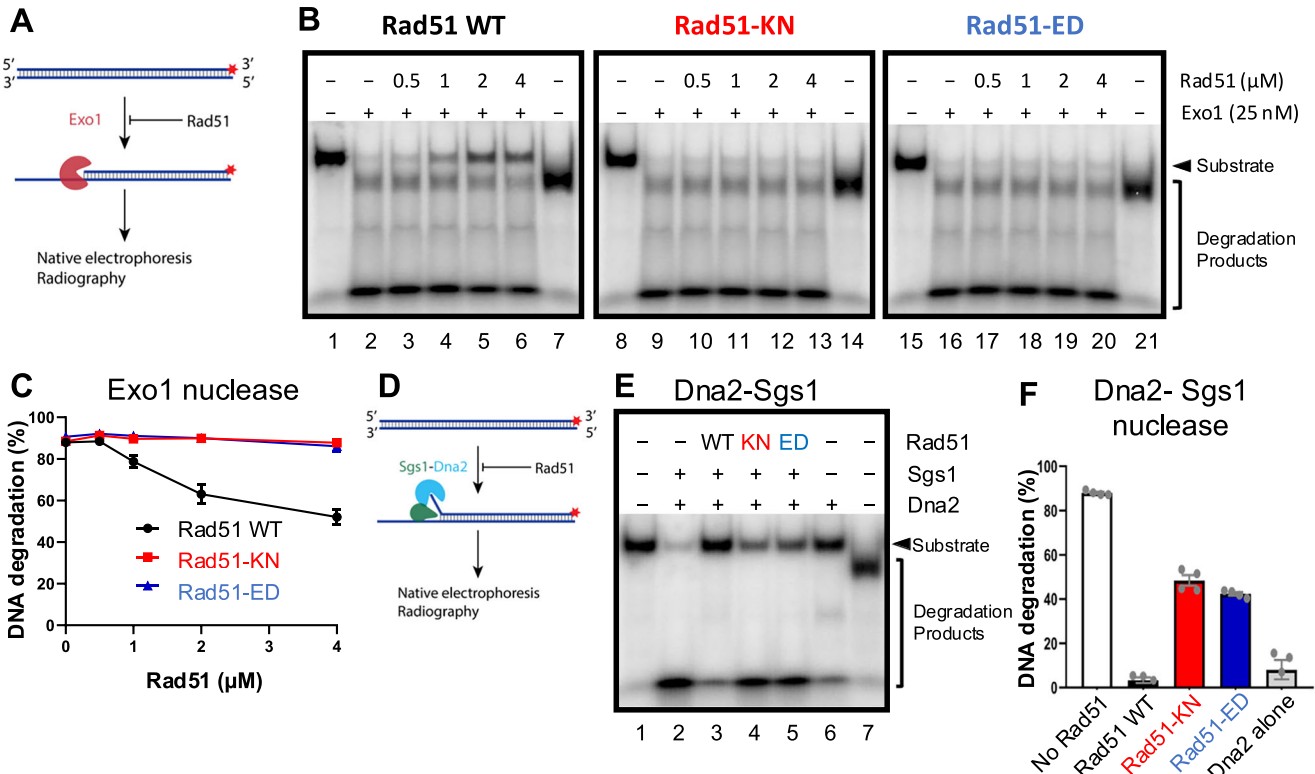

**Fig. 6 | Rad51-KN and Rad51-ED are defective in protecting dsDNA from degradation by Exo1 or Dna2-Sgs1. A** Cartoon depicting the nuclease assay with Exo1 performed in panels B-C. The red asterisks represent the position of the radioactive label. **B** Representative nuclease assays of Exo1 using a 100 bp DNA substrate in the presence of increasing concentrations of Rad51 wild type and variants, as indicated. In lanes 7, 14 and 21, the substrate was boiled to indicate the position of the ssDNA. **C** Quantitation of experiments such as shown in (**B**). Averages shown; error bars, SEM; 3 independent experiments ($n = 3$). **D** Cartoon depicting the nuclease assay with Sgs1-Dna2 performed in (**E**, **F**). The red asterisks represent the position of the radioactive label. **E** Representative nuclease assay with Dna2-Sgs1 in the presence of 4 μM Rad51 wild type and variants. In lane 7, the substrate was boiled to indicate the position of the ssDNA. **F** Quantitation of experiments such as shown in E. Averages shown; error bars, SEM; 3 independent experiments ($n = 4$). Source data are provided as a Source Data file.

(hypothetically fork regression) similar to the situation in cells with a complete HR defect. The *rad51-KN* mutant shows a less accentuated phenotype.

Furthermore, we tested a PCNA mutant (encoded by the *POL30* gene in budding yeast), which eliminated the K127 and K164 acceptor sites (*pol30-K127R,K169R* abbreviated as *pol30-KKRR*) for ubiquitylation and sumoylation to obstruct Srs2-mediated anti-recombination, TLS, and TS (likely fork regression) pathways. This double mutant displayed only very mild MMS sensitivity at the low (0.0033%) concentration used (Fig. 7F row 3), as expected[41]. This result suggests that under these conditions the salvage HR is the predominant pathway. This notion is confirmed by the extreme synergy in MMS sensitivity with an HR defect as well as the strong growth/survival defect on YPD (Fig. 7F, compare *pol30-KKRR* row 3 with *rad54Δ pol30-KKRR* row 4 or *rad51Δ pol30-KKRR* row 7 or *rad51Δ rad54Δ pol30-KKRR* row 8). *rad51-ED* mimicked again the behavior of a complete HR defect (Fig. 7F; compare *rad51-ED* row 9 with *rad51-ED pol30-KKRR* row 11) including the remarkable growth/survival defect on YPD lacking MMS. Consistent with the genetic interaction analysis involving *rev3Δ* and *mms2Δ*, the *rad51-KN* mutant shows a less pronounced phenotype which was enhanced by a complete HR defect (Fig. 7G, compare *rad51-KN* row 9 with *rad51-KN pol30-KKRR* row 11 and with *rad51-KN rad54Δ pol30-KKRR* row 12). We conclude that in *rad51-ED* and *rad51-KN* cells, HR-independent template switching and TLS process a larger fraction of MMS-stalled forks than in wild type cells.

Finally, we used a subtle mutation in *RAD5*, *rad5-G535R*, which by itself has little to no phenotype[71] (Supplementary Fig. 3C). Rad5 promotes PCNA polyubiquitylation and template-switching and also plays a role in recruiting TLS polymerases, affecting two major post-

replication repair pathways. In consequence, the deletion of the *RAD5* gene causes one of the most pronounced MMS sensitivities caused by a single mutant in budding yeast, requiring a *rad5* allele with a more subtle phenotype. In serial dilution assays, we noticed some suppression of the MMS sensitivity of *rad54Δ* by *rad51-ED* and *rad51-KN* which depended on the wild type *RAD5* gene (Supplementary Fig. 3B compare *rad51-ED rad54Δ RAD5* row 4 with *rad51-ED rad54Δ rad5-G535R* row 3 and *rad51-KN rad54Δ RAD5* row 10 with *rad51-KN rad54Δ rad5-G535R* row 11). The discrepancy between the results from the serial dilution (Supplementary Fig. 3) and colony formation assays regarding *rad54* suppression (Fig. 1) is explained by the fact that serial dilution assays measure a sum of growth and survival, whereas colony formation measures survival only. These results reinforce the conclusion that *rad51-ED* and *rad51-KN* rely more heavily on TLS and HR-independent template switching to process stalled replication forks. The *rad5-G535R* mutation did not affect the IR sensitivity of *rad51-ED* or *rad51-KN* (Supplementary Fig. 3A) and has been described as wild type for IR resistance (https://wiki.yeastgenome.org/index.php/CommunityW303.html).

From this genetic analysis with four genes affecting single or multiple post-replication repair pathways, we conclude that the balance between post-replication repair pathways is significantly altered in the *rad51-ED* and *rad51-KN* mutants.

### *rad51-ED* and *rad51-KN* display a strong mutator phenotype

HR-defective cells such as *rad51Δ* or *rad54Δ* mutants display a considerable mutator phenotype likely caused by spontaneous DNA damage that, instead of being repaired or tolerated by high fidelity HR, is processed by the error-prone TLS pathway, causing an increase in

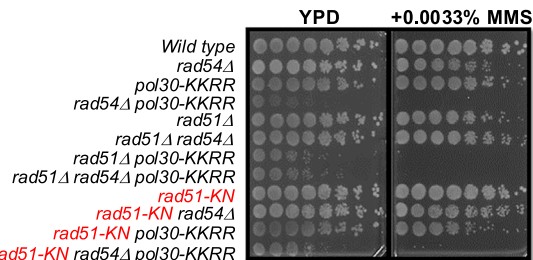

**A.** *rev3* *rad51-ED*

**B.** *rev3* *rad51-KN*

**C.** *mms2 rad51*

**D.** *mms2* *rad51-ED*

**E.** *mms2* *rad51-KN*

**F.** *pol30-KKRR* *rad51-ED*

**G.** *pol30-KKRR* *rad51-KN*

**Fig. 7 | rad51-ED and rad51-KN display strong synergy with defects in post-replication repair. A**, **B** Analysis with *rev3*. Serial dilutions of wild type (WDHY1636), *rad54Δ* (WDHY1275), *rev3Δ* (WDHY3584), *rad54Δ rev3Δ* (WDHY3588), *rad51Δ* (WHDY2542), *rad51Δ rad54Δ* (WDHY2544), *rad51Δ rev3Δ* (WDHY3787), *rad51Δ rad54Δ rev3* (WDHY3788), *rad51-ED* (WDHY3548), *rad51-ED rad54Δ* (WDHY3546), *rad51-ED rev3Δ* (WDHY3784), *rad51-ED rad54Δ rev3Δ* (WDHY3785), *rad51-KN* (WDHY3962), *rad51-KN rad54Δ* (WDHY3572), *rad51-KN rev3Δ* (WDHY3589), *rad51-KN rad54Δ rev3Δ* (WDHY3789). **C**–**E** Analysis of survival following exposure to MMS in the following genotypes: wildtype (WT) (WDHY3960), *rad51Δ* (WDHY3898), *rad54Δ* (WDHY3961), *mms2Δ* (WDHY3855), *rad51Δ mms2Δ* (WDHY3952), *rad54Δ mms2Δ* (WDHY3956), *rad51-KN* (WDHY3962), *rad51-KN mms2Δ* (WDHY3958), *rad51-KN rad54Δ mms2Δ* (WDHY4564), *rad51-ED*

(WDHY3548), *rad51-ED mms2Δ* (WDHY5169), *rad51-ED rad54Δ mms2Δ* (WDHY6123). The mean and SEM were calculated from a minimum of three independent cultures (*n* = 3) with some MMS concentrations and strains up to *n* = 6 (see Source Data). **F**, **G** Analysis with *pol30-KKRR*. Serial dilutions of wild type (WDHY1636), *rad54Δ* (WDHY1275), *pol30-KKRR* (WDHY3555), *rad54Δ pol30-KKRR* (WDHY3563), *rad51Δ* (WHDY2542), *rad51Δ rad54Δ* (WDHY2544), *rad51Δ pol30-KKRR* (WDHY3783), *rad51Δ rad54Δ pol30-KKRR* (WDHY3780), *rad51-ED* (WDHY3548), *rad51-ED rad54Δ* (WDHY3546), *rad51-ED pol30-KKRR* (WDHY3561), *rad51-ED rad54Δ pol30-KKRR* (WDHY3570), *rad51-KN* (WDHY3962), *rad51-KN rad54Δ* (WDHY3572), *rad51-KN pol30-KKRR* (WDHY3557), *rad51-KN rad54Δ pol30-KKRR* (WDHY3568). Source data are provided as a Source Data file.

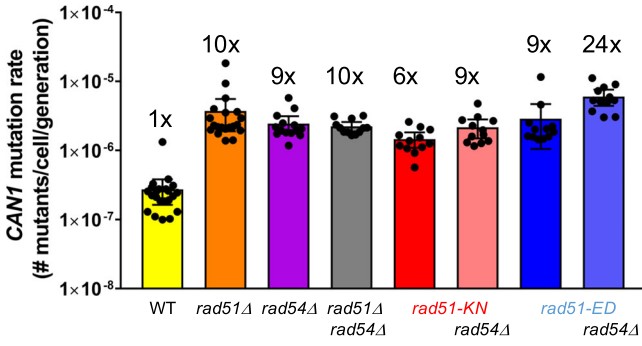

**Fig. 8 | rad51-ED and rad51-KN display a strong mutator phenotype.** The spontaneous *CAN1* mutation rate from a minimum of 12 single clones was assayed. Freshly dissected diploid strains allowed for multiple strains of the following genotypes to be created: WT (*n* = 22) and *rad51Δ* (*n* = 20) (WDHY4204), *rad54Δ* (*n* = 14) (WDHY4205), *rad51Δ rad54Δ* (*n* = 12) (WDHY3162), *rad51-KN* (*n* = 12) (WDHY4208), *rad51-KN rad54Δ* (*n* = 12) (WDHY4210), *rad51-ED* (*n* = 12) (WDHY4207), and *rad51-ED rad54Δ* (*n* = 12) (WDHY4209). The mean rate and ± 95% confidence interval were calculated for each genotype. Fold increase compared to WT is given. Source data are provided as a Source Data file.

spontaneous mutagenesis. We used the forward mutagenesis *CAN1* assay to determine the rate of spontaneous mutations and saw the expected 9-10-fold increase in *rad51Δ*, *rad54Δ* single and double mutants (Fig. 8). It was previously reported that this increase largely depends on the central TLS polymerase component Rev3[72]. To our surprise, *rad51-ED* and *rad51-KN* showed a significant increase of 9-fold and 6-fold, respectively, in their mutation rate. These values are not significantly different from those of the *rad51Δ*, *rad54Δ* single and double mutants. Both mutants did not suppress the mutator phenotype of the *rad54Δ* mutant (Fig. 8). Instead, *rad51ED* slightly elevated the mutation rate in *rad54Δ* cells from 9-fold in the single mutants to 24-fold in the *rad51-ED rad54Δ* double mutant, which may suggest that the *rad51-ED* mutant favors mutagenic events that are suppressed by *RAD54*. We conclude that *rad51-ED* and *rad51-KN* show a significant mutator phenotype due increased engagement of TLS at stalled replication forks.

## Dramatic shift from template switching to TLS

To investigate the effect of *rad51-ED* and *rad51-KN* mutants on lesion bypass, we took advantage of the iDamage approach (Fig. 9A)[50]. In short, a plasmid containing the single lesion of interest is stably inserted at a specific locus of the yeast genome by means of the Cre recombinase and modified lox sites. Cre-mediated recombination between the mutant lox71 (on the vector) and the mutant lox66 (in the genome) leads to a double mutant lox site that is no longer recognized by the Cre recombinase, avoiding excision of the lesion-containing vector. Lesion bypass is monitored by counting blue and white colonies, as the lesion is inserted in the *lacZ* reporter gene (Fig. 9A). While this system does not allow to study MMS-induced DNA damage, we investigated the bypass of two common lesions: the thymine-thymine pyrimidine(6−4)pyrimidone photoproduct [TT(6−4)], and the N2-dG-Acetylaminofluorene (N2dG-AAF). All experiments were performed in a parental strain where NER and mismatch repair are inactivated to prevent the repair of the lesion and the DNA loop used as a strand marker, respectively. We measured a level of TLS of ~5% and ~17% for the TT(6−4) and the G-AAF lesions in this parental strain (Fig. 9B). The inactivation of *rad51* led to a decrease in the level of the template-switching pathways (HR-mediated gap repair, fork regression) and a concomitant increase in the level of TLS (Fig. 9B). In *rad51Δ* cells about half of the template switching events remained, suggesting a significant contribution of Rad51-independent template switching to the total number of tolerance events. This reflects the competition

between template switching and TLS: when a template switching pathway is inhibited, TLS is favored. Upon introduction of the same lesion in the *rad51-ED* and *rad51-KN* mutants, we observed the same decrease in the use of template switching and the same increase in TLS (Fig. 9B).

## Defects in binding to stalled forks and forming sister chromatid junctions

The recombinational processing of MMS-induced DNA lesions is associated with the formation of Rad51-dependent sister-chromatid junctions (SCJs). The previous genetic results suggested that there is a defect in *rad51-ED* and *rad51-KN* cells to process MMS-stalled replication forks. The accumulation of these recombination intermediates can be detected as X-shaped structures by 2D electrophoresis in *sgs1Δ* cells, which are defective in their dissolution[73]. To address the functionality of Rad51-ED and Rad51-KN in the recombinational response to MMS, we measured the accumulation of SCJs in cells synchronized in G1 and released into S phase in the presence of MMS (Fig. 10A). Rad51-ED and Rad51-KN are affected in their ability to promote SCJs as inferred from the behavior of the mutants. *sgs1Δ rad51-KN* cells displayed a 1.7-fold reduction in the amount of SCJs along the analyzed times that was mostly due to a delay in the accumulation of SCJs as compared to the wild type. Remarkably, *sgs1Δ rad51-ED* was severely affected in the accumulation of SCJs and reduced to ~20% of the wild type levels. We conclude that the *rad51-ED* and, to a lesser degree, *rad51-KN* mutants are defective in HR at MMS-stalled replication forks leading to increased usage of alternate post-replication repair pathways. Alternatively, it is possible that in these mutants such structures are less stable or more likely to be degraded.

The pronounced MMS sensitivity (Fig. 1, Supplementary Fig. 1) and a severe defect in SCJs formation (Fig. 10A) of *rad51-ED* cells contrasts with its near wild type behavior in direct genetic assays of HR (Fig. 2). A mechanistic difference between the roles of Rad51 in HR at DSBs and MMS-induced ssDNA is that the role of Rad51 at ssDNA lesions is coupled to replication[35], whereas the role of Rad51 at DSBs is replication-independent[74,75]. Thus, Rad51-ED might be specifically affected in the binding to MMS-induced lesions during S-phase. To address this, we constructed chimeras of Rad51 with the MNaseI (Rad51-MN) and followed their binding to DNA lesions by Chromatin Endogenous Cleavage (ChEC) analysis. In this approach, the induction of the nuclease activity of the chimera by treating cells with Ca²⁺ will only generate a detectable cut in the DNA if the repair protein is targeted to a lesion that is not a DSB. Thus, we can infer Rad51 binding to DNA if DNA is digested upon Ca²⁺ treatment[35]. In the absence of MMS, most DNA remained as a single, high molecular DNA band on top of the gel; in the presence of MMS both Rad51-MN and Rad51-KN-MN generated a smear below the top band, whereas Rad51-ED-MN hardly digested the top band (Fig. 10B). These results were fully reproducible in two additional independent experiments (Supplementary Fig. 11A). We conclude that Rad51-ED is severely impaired in binding to MMS-stalled replication forks or that its DNA binding substrate is not formed or degraded. Control experiments showed that the Rad51-ED-MN fusion did not aggravate the DNA damage sensitivity of the *rad51-ED* mutant (Supplementary Fig. 11B), suggesting that the observed defect is not a consequence of the MN fusion. The Rad51-KN-MN fusion sensitized the *rad51-KN* mutant (Supplementary Fig. 11B), although the ChEC assay showed no defect for this mutant.

## No effect on interaction with Rad55-Rad57

Rad51 paralog complex Rad55-Rad57 functions in the nucleation of Rad51 filaments on RPA-coated ssDNA[76]. It also stabilizes the Rad51-ssDNA filament against dissociation by the Srs2 helicase that has the ability to strip Rad51 from ssDNA, therefore acting as an anti-recombinase[56–58]. Detailed genetic analysis showed a strong similarity between the phenotypes of cells lacking Rad55 and the *rad51-ED* or

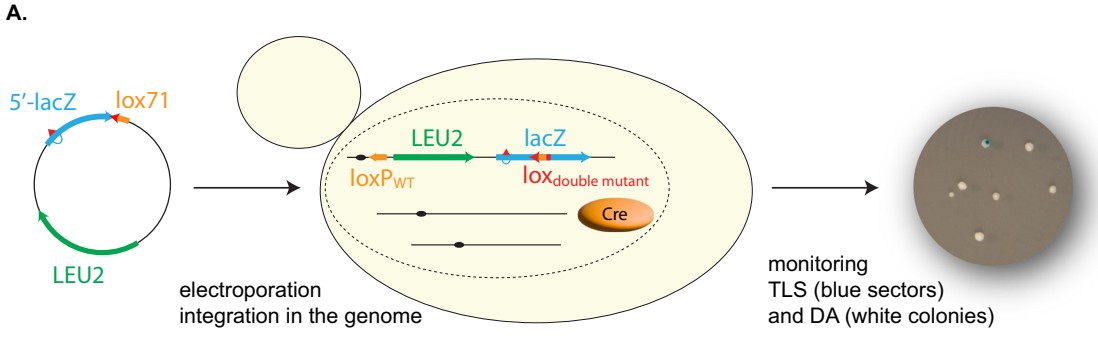

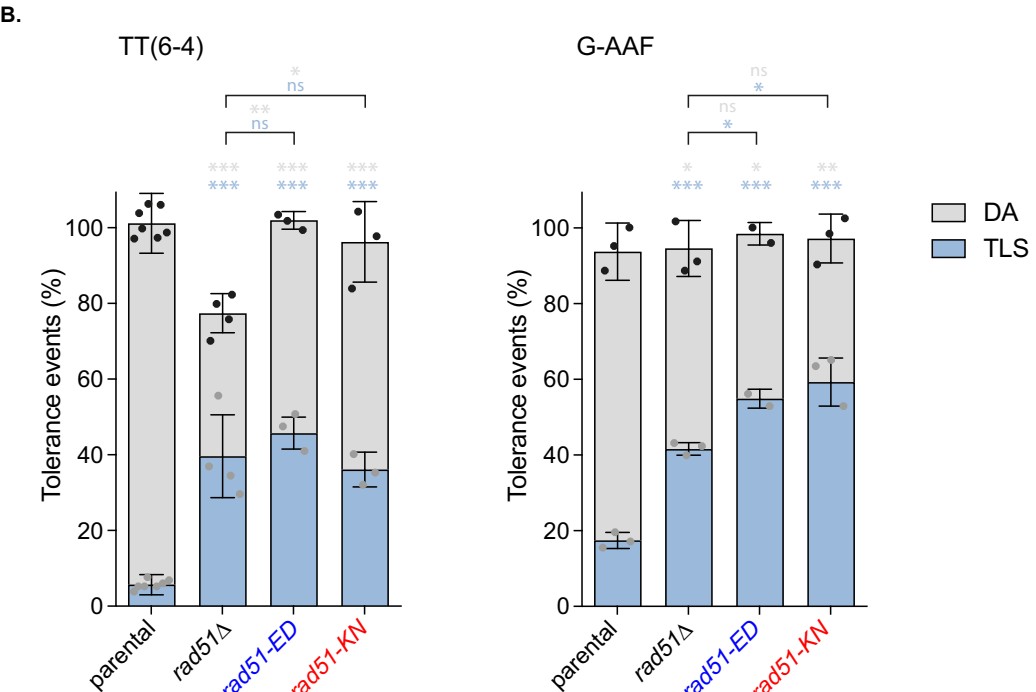

**Fig. 9 | rad51-ED and rad51-KN shifts the balance from damage avoidance to translesion synthesis in post-replication repair. A** Outline of the iDamage approach: a non-replicative plasmid containing a single lesion is integrated into one of the yeast chromosomes using Cre/lox site-specific recombination. Using two lox mutants (lox71 on the integrating vector and lox66 on the genome (not shown)), the integration event leads to a lox site with two mutation that prevent excision of the integrated vector indicated as lox_{double mutant}. The integrative vector carrying a selection marker (*LEU2*) and the 5′-end of the *lacZ* reporter gene containing a single lesion is introduced into a specific locus of the chromosome with the 3′-end of *lacZ*. The precise integration of the plasmid DNA into the chromosome restores a functional *lacZ* gene, enabling the phenotypical detection of TLS and DA events (as blue and white colonies on X-gal indicator media). **B** Partitioning of DNA damage tolerance pathways at a TT(6–4) photoproduct and at a G-AAF lesion. Tolerance events represent the percentage of cells able to survive in the presence of the integrated lesion compared to the lesion-free control in wild type (*n* = 7 TT(6–4), *n* = 3 IAF) (SC53, SC55), *rad51Δ* (*n* = 4 TT(6–4), *n* = 3 IAF) (SC253, SC255), *rad51-ED* (*n* = 3 TT(6–4), *n* = 2 IAF) (SC844, SC845) and *rad51-KN* (*n* = 3 TT(6–4), *n* = 3 IAF) (SC868, SC869) strains. The data represent the average and standard deviation. N values are given for each genotype and lesion. Unpaired two-tailed t-test was performed to compare TLS and DA values from the different mutants to the parental strain. *P < 0.05; **P < 0.005; ***P < 0.0005 indicated for TLS (light blue) and DA (gray), ns not significant. Source data are provided as a Source Data file.

*rad51-KN* mutants described here in their dependence on the TLS polymerase Rev3[77]. This opened the possible interpretation that the *rad51-ED* and *rad51-KN* mutants affected the interaction of Rad51 with the Rad55-Rad57 complex. Using purified wild-type Rad51 as well as Rad51-ED and RAD51-KN proteins in pull-down experiments with the purified Rad55-Rad57 complex, we could demonstrate that both mutants interacted normally with the Rad51 paralog complex (Supplementary Fig. 12). We conclude that the phenotypes of the *rad51-ED* and *rad51-KN* mutants is not caused by a defect in interacting with Rad55-Rad57.

## Discussion

The major conclusion of this work is that properties of Rad51 determine pathway usage at stalled replication forks. We have isolated mutants in *RAD51, rad51-ED and rad51-KN* that profoundly shift the balance in the usage of post-replication repair pathways to TLS and fork regression at replication forks stalled by MMS (Supplementary Fig. 12A). This leads to a reliance of genes controlling TLS and fork regression (Fig. 7, Supplementary Fig. 3), a strong mutator phenotype (Supplementary Fig. 8), a directly demonstrated shift from HR to TLS in post-replication repair (Fig. 9), and a defect targeting Rad51 to

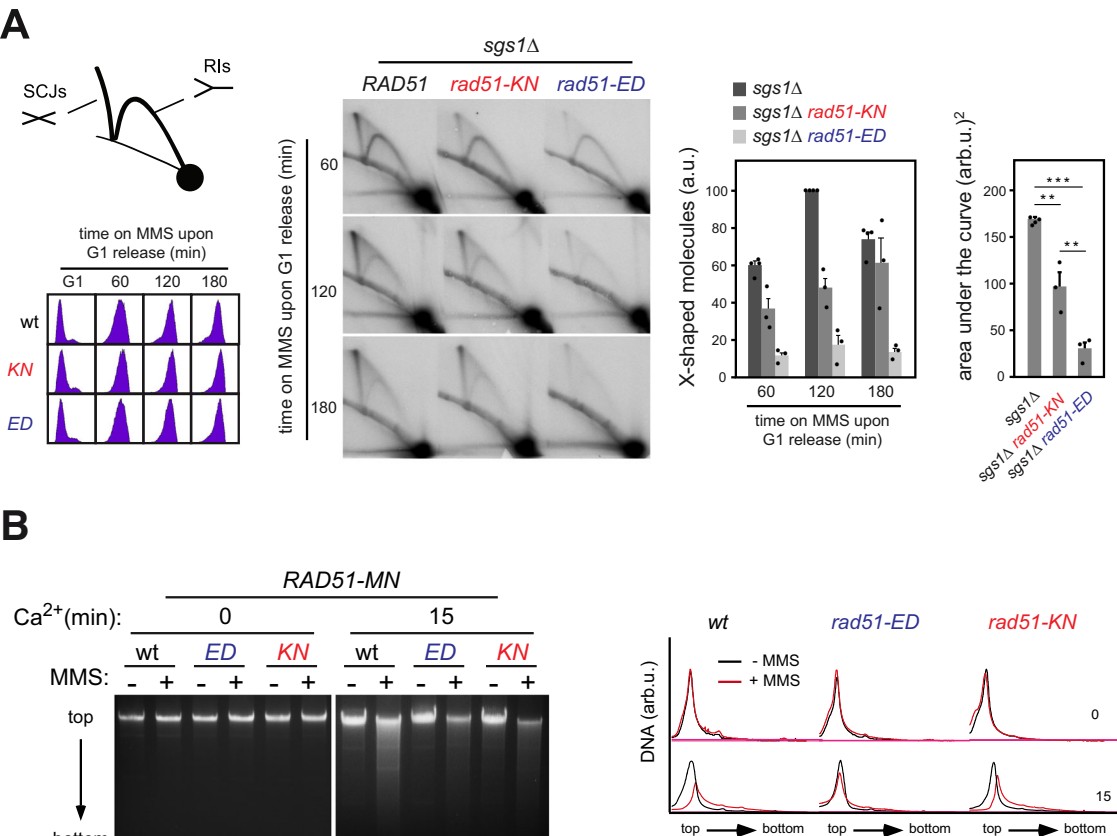

**Fig. 10 | Rad51-ED is severely affected in binding to damaged DNA and Sister Chromatid Junction formation. A** *rad51-KN* and *rad51-ED* are defective in the accumulation of Sister Chromatid Junctions (SCJ), as determined by 2D gel analysis of X-shaped molecules in *sgs1Δ* (W303sgs1), *sgs1Δ rad51-KN* (W303sr305) and *sgs1Δ rad51-ED* (W303sr135) cells synchronized in G1 and released in the presence of 0.033% MMS. A schematic representation of the migration pattern of X-shaped (SCJs) and Y-shaped (replication intermediates) molecules, as well as the DNA content analysis of the different cultures along the time course, is shown on the left. The panel on the right shows the amounts of X-shaped molecules relative to the total amount of molecules at the indicated times. The highest value is set at 100, and the mean and SEM from four independent time courses are given. The area under the curve (AUC) is also plotted. Two and three asterisks indicate statistically significant differences according to a One-Anova (Bonferroni) test (*P* < 0.001 and 0.0001, respectively). **B** Rad51 binding to replicative DNA lesions as determined by ChEC analysis of exponentially growing *RAD51-MN* (wt) (wR51MN-2), *rad51-ED-MN* (wR51-135MN) *and rad51-KN-MN* (wR51-305MN) cells incubated with and without 0.05% MMS for 2 h. Total DNA from cells permeabilized and treated or not with $Ca^{2+}$ for 15 min is shown, as well as the DNA digestion profiles. ChEC experiments were repeated three times with similar results. Source data are provided as a Source Data file.

stalled forks and generating sister chromatid joints (Fig. 10, Supplementary Fig. 11B). The biochemical analysis showed a major defect of the mutants in binding dsDNA (Fig. 4, Supplementary Figs. 6, 7), which led to a loss of protection from exonucleases (Fig. 6, Supplementary Fig. 10). The severity of the defects in vivo and in vitro correlated with Rad51-ED being more affected than Rad51-KN, suggesting that the dsDNA binding is a major reason, but possibly not the only reason, for the observed in vivo phenotypes. Our conclusion that properties of Rad51 determine post-replication repair pathway usage is further supported by previous studies showing that cells lacking Rad55 show elevated mutation rates and strong dependence on alternative post-replication repair pathway controlled by *REV3* and *MMS2*[77–80], similar to the Rad51 mutants isolated here. The Rad55-Rad57 complex stabilizes Rad51 filaments[58], but the mechanisms involved are likely different as the Rad51 mutants showed no defect in Rad55 interaction (Supplementary Fig. 12). In sum, the combined data show that Rad51 filament properties affect pathway usage in post-replication repair.

*What properties of Rad51 determine the phenotype of the ED and KN mutants?* The key difference emanating from the biochemical analysis was the defect of the mutants to bind dsDNA (Fig. 4, Supplementary Figs. 6, 7 The much stronger defect of the Rad51-ED mutant protein than the Rad51-KN mutant in binding to dsDNA correlated with their in vivo phenotypes, where the *rad51-ED* mutant consistently showed a stronger phenotype than the *rad51-KN* mutant. Hence, we suggest that

a key determinant of the ED and KN mutant phenotypes is the deficit in binding dsDNA. The ability to bind dsDNA with relatively high affinity is a newly acquired property in eukaryotic Rad51 proteins which is lacking in bacterial RecA[16–20]. These observations suggest that the ability of Rad51 to bind dsDNA is unrelated to its HR function. Recent biochemical evidence with human RAD51 supports the model that dsDNA binding by RAD51 is critical for its function in fork protection against nucleolytic degradation[31,32]. The Rad51-ED and RAD51-KN mutants reinforce this model as the mutant proteins show a defect in protecting dsDNA from exonucleolytic attack (Fig. 6, Supplementary Fig. 10).

*Where is RAD51 binding dsDNA at stalled forks?* Fork stalling in budding yeast primarily results in non-regressed forks. Extensive fork regression was only observed in cells defective for DNA damage checkpoints[45]. This contrasts with the situation in mammalian cells, where fork regression appears to be a primary response to fork stalling[46]. Stalled forks have several ssDNA-dsDNA transitions that could be sensitive to nuclease degradation (Supplementary Fig. 13B left). In addition, a recent study proposed a function of RAD51 in fork stabilization by binding to dsDNA to lock in the CMG helicase and prevent loss of the replicative helicase to enable direct restart[32]. Interestingly, Rad51 was found to physically interact with the MCM complex[81], providing a potential mean to nucleate Rad51 filaments at stalled forks (Supplementary Fig. 13B). It is unclear whether each of these potential dsDNA binding sites of RAD51 is relevant for fork

protection, but the multitude of nucleases involved, including EXO1, DNA2, MRE11, and potentially others[29,82], may suggest that several sites may be relevant depending on context. We speculate that in the *rad51-ED* and *rad51-KN* mutants, DNA degradation would convert internal gaps (Supplementary Fig. 13B, left) and regressed forks (Supplementary Fig. 13B, right) into gapped forks that would be extended by TLS to restart replication.

Our results also provide indirect evidence for an HR (Rad51)-independent template switching mechanism in budding yeast that depends on PCNA-poly ubiquitylation (Mms2), which we suspect is fork regression. First, template switching is not eliminated in *rad51Δ* cells (Fig. 9). This suggests the existence of a Rad51-independent template switch mechanism, which could be fork regression driven by motor proteins such as Rad5. Second, *rad51Δ* cells show additive MMS-sensitivity with Mms2 (Fig. 7C–E). Mms2 functions through K63 poly-ubiquitylation of PCNA on K164[41,83]. This result suggests that Mms2 controls an HR-independent template switch process in postreplication repair, likely fork regression as gap repair by TS is Rad51 dependent. Third, in *ubc13* cells where PCNA polyubiquitination is prevented, template switching is only partially eliminated as in *rad51Δ* cells[50,84] (Fig. 9). Together these data strongly suggest the existence of two template switching pathways, one Rad51-dependent (gap repair) and one Rad51-independent (likely fork regression) (Supplementary Fig. 13A). Fourth, in checkpoint-deficient yeast cells (*rad53*) regressed forks readily form[45] and likely present a target for dsDNA-specific exonucleases like Exo1 with likely deleterious consequences (Supplementary Fig. 13B right). Rad51 binding to dsDNA in these regions is expected to suppress nucleolytic attack. In mammalian cells, motor proteins implicated in fork regression (SMARCAL1, ZRANB3, and HLTF) facilitate DNA degradation, suggesting that regressed forks are substrates for DNA degradation[85–87]. In fact, it was found that eliminating Exo1 effectively suppresses the MMS sensitivity of yeast *rad53* mutants[88]. While regressed forks might be rare or highly unstable in wild type cells, there are conditions, such as in checkpoint defective cells, where fork regression is readily documented in budding yeast[45]. This implies that the mechanisms for fork regression exist in budding yeast but are under negative control. Hence, Rad51 may also be binding to regressed forks to protect against dsDNA nucleases (Supplementary Fig. 13B right).

The *rad51-ED* and *rad51-KN* mutants are proficient in inter-sister recombination, only *rad51-ED* shows a small reduction in spontaneous events (Fig. 2), which is consistent with the ability of the mutant proteins to form D-loops in vitro (Fig. 5). However, the mutants are defective in forming SCJs (Fig. 10) and inter-chromosomal recombination, consistent with a strong defect to form D-loops in vivo (Fig. 3). We suspect this difference is a reflection of the defect to form extended nucleoprotein filaments in vitro (Supplementary Fig. 7) and in vivo (Supplementary Fig. 8). Similar to rad51-ED and rad51-KN, the rad51-II3A mutant forms short filaments that cannot be visualized by IF[89]. However, in contrast the Rad51-II3A protein is defective in D-loop formation and the mutant is as sensitive to IR as the rad51 deletion mutant[89], unlike rad51-ED and rad51-KN.

In some respects, Rad51-ED and Rad51-KN are similar to the human RAD51-S181P mutation that lacks the interaction with the BRCA2 exon27-encoded RAD51 filament interaction site[90]. Like RAD51-S181P, Rad51-ED and RAD51-KN are proficient in D-loop formation in vitro and inter-sister recombination in cells, but defective in fork protection. It is not known if RAD51-S181P is proficient in genome-wide homology search or whether the protein has a defect in binding dsDNA. The distinct chromosomal aberration profile of cells with RAD51-S181P compared to the ATPase deficient RAD51-KR mutant may suggest that RAD51-S181P affects pathway usage at stalled replication forks, although this was not directly studied. The common denominator for the mutants is that Rad51 filament properties affect its function in fork protection.

It has previously been noted that genome-wide homology has different requirements and characteristics than local homology search. The context-specific requirement for long range resection in genome-wide homology search involves activation of the DNA damage signaling and can be overcome by Rad51 overexpression[91]. It is likely that both mechanisms affect Rad51 filament length and quality, as the Rad51 paralogs Rad55 and Rad57 are direct targets of the Mec1 and Rad53 DNA damage signaling kinases[92–94] and absence of the paralogs causes a similar defect as rad51-ED and rad51-KN[77]. Moreover, in vivo monitoring of homology search revealed that the initial search is conducted by short Rad51-ssDNA filaments, whereas genome-wide search requires a transition to more extensive nucleoprotein filaments[95]. We suspect that SCJs and inter-chromosomal recombination require longer Rad51-ssDNA filaments, while DSB repair including the type of Rad51-mediated template switch started at regressed forks (Supplementary Fig. 13A) can be accomplished by shorter filaments.

In sum, the ability of Rad51 to bind dsDNA with relatively high affinity appears to correlate with its acquired function to protect stalled replication forks against nucleolytic degradation and potentially retain the replicative helicase for direct restart. The Rad51-ED and Rad51-KN mutants show that properties of Rad51 determine pathway usage in post-replication repair which likely involves its ability to bind dsDNA and form extended nucleoprotein filaments.

## Methods

### Yeast strains and growth conditions

Standard techniques for yeast growth and genetic manipulation were used[96]. All yeast strains used for genetic experiments are in the W303-1A background and WT *RAD5* (Supplementary Table 1), except for some strains used in Supplementary Fig. 1C which are *rad5-G535R* as noted. See Supplementary Table 2 for all plasmids used. See Supplementary Table 3 for all oligonucleotides and their sequences.

### Rad51 mutant isolation and genomic integration

To screen for *rad51* separation-of-function mutants, a random pool was created by mutagenic PCR using the GeneMorph II EZClone Domain Mutagenesis Kit that employs two error prone polymerases to achieve balanced mutations in all four bases. The *RAD51* open reading frame is 1.2 kb in length; the ideal pool of mutants would have one to a few mutations per gene. Mutagenic PCR conditions were empirically optimized to obtain a mutation rate of ~1 mutation per kb, by varying the amount of template DNA and reducing the number of PCR cycles using olWDH632 and olWDH633 which are 400 nt upstream and downstream of the *RAD51* open reading frame, respectively, and plasmid pWDH957, which contains the wild type *RAD51* gene with 1000 bp upstream and downstream of the *RAD51* open reading frame cloned as a 3.2 kb *Hind*III-*Sac*I fragment into Yep351 (pWDH958) using olWDH830 and olWDH1355. The pool used averaged two mutations per gene. The screen was conducted in a *rad51Δ rad54^{ts}* strain (WDHY2546) to take advantage of the conditional phenotype. The *rad54^{ts}* allele was determined to be caused by a single amino acid change (C692Y), which leads to loss of protein at the restrictive temperature. The mutagenized *RAD51* pool was co-transformed with pWDH957 digested with *Stu*I and *Nru*I to cut out almost the entire RAD51 ORF from −22 to 47 bp upstream the stop codon for in vivo recombination to form functional circular YEp351 plasmids containing a mutagenized *RAD51* gene. Approximately 2000 transformants were screened. Of these, 40 candidates exhibited improved viability when challenged with MMS compared to the *rad51Δ rad54^{ts}* strain at 37 °C. The plasmid dependence of the phenotype was confirmed. Plasmids were recovered in *E. coli*, the *RAD51* sequence was established, and the phenotype re-confirmed by transformation of a host strain that has never been exposed to MMS.

The mutations *rad51-ED* and *rad51-KN* were transferred to the native *RAD51* locus of WDHY2217 that contained a complete deletion

of the *URA3* gene to preclude reversion events using the PCR-based allele replacement method of Erdeniz et al.[97] with the *RAD51* primers olWDH1351 and 1353 as well as the *K. lactis URA3* primers olWDH1357 and 1358. Correct integration and the integrity of the integrated sequences were confirmed by amplifying the *RAD51* locus and sequencing the entire gene on both strands.

## Serial dilution assays

The serial dilution spot assays were performed on solid YPD or YPD with the addition of MMS at the dilutions specified. To assay for sensitivity to ionizing radiation (IR), YPD plates were spotted with cells and allowed to dry before being exposed to IR (Cesium-137 source). Cells were spotted at 5-fold dilutions starting at approximately $2 \times 10^7$ cells/ml (OD600 = 1). The plates were incubated at 30 °C and photographed daily for up to 4 days.

## Immunoblots

Lysates were prepared by trichloroacetic acid (TCA) precipitations from cells grown to early log phase. Cells were re-suspended in 20% TCA before cell disruption utilizing a FastPrep machine (FP120, Bio101, Savant) on setting "4.5", four times for 45 seconds. Beads were washed with 5% TCA. Precipitated protein was collected at $21,100 \times g$ in an Eppendorf centrifuge for 10 min, resuspended in 2x Laemmli sample buffer[98], and neutralized with Tris-base before analysis by gel electrophoresis. Immunoblots were conducted as described[92]. Lysate from 0.5 OD600 units of cells were loaded and separated on a 10% SDS-PAGE gel before transfer to PVDF membrane. The membrane was blocked in 5% skim milk dissolved in TBST (10% Tris pH 7.5, 150 mM NaCl, 0.1% Tween-20) and incubated with either anti-Rad51 (1:1000, sc-133089, Santa Cruz Biotechnology) or anti-GAPDH (1:5000, GA1R, Thermo Scientific) antibodies. Membranes were developed by chemiluminescence (Clarity, Bio-Rad) and visualized using a GE Amersham Imager 600. All bands were quantified through densitometry using Image Lab (Bio-Rad).

## Immunofluorescence

Rad51 foci were detected by immunofluorescence as described[99]. In short, log-phase cells (5 ODU) were crosslinked in 2% formaldehyde (37% stock) for 30 min at 30 °C shaking. Cells were harvested at 800xg for 5 min in a 15 ml conical tube and rinsed once in 1.0 ml 1 M potassium phosphate pH 6.5. The supernatant was removed, and cells were resuspended in 1.0 ml of solution P (0.1 M potassium phosphate pH 6.5, 1.2 M sorbitol), transferred to a microfuge tube and incubated with 15 μl zymolyase (10 mg/ml) in solution P and 5 μl β-mercaptoethanol for 30 min at 30 °C shaking. While the cells were incubating, slides were prepared by overlaying 200 μl 0.3% poly-lysine (in ddH₂O) for 15–30 min followed by a rinse with ddH₂O. Spheroblasts were harvested at $800 \times g$ in a microfuge for 3 min, resuspended in 1–2.0 ml of solution P at -0.5–1.0 ODU/ml and a minimum of 250 μl cells were incubated on each slide for 15–30 min. Cells were fixed by adding 100% ice cold MeOH drop wise onto the cell suspension. Slides were placed into a Coplin jar containing ice cold MeOH and incubate at −20 °C for a minimum of 20 min. Slides were air dried and rehydrated in 1 x PBS (phosphate-buffered saline: 137 mM NaCl, 2.7 mM KCl, 8 mM Na₂HPO₄, and 2 mM KH₂PO₄.) and then overlaid with 200 μL anti-Rad51 antiserum (1:100)[92] in 3% BSA in PBS and incubated for 1 h at 37 °C. The slides were washed twice in PBS/0.1% Tween 20 for 3 min each then 1x in PBS alone for 3 min while shaking. The slides were overlain with 200 μL goat anti-rabbit antibody conjugated to Alexa 488 (1:1,000; AbCam ab150077) in 3% BSA/PBS and incubated for 30 min at 37 °C followed by the washing scheme outlined above. The slides were then stained for 1 min in DAPI (1ug/mL))/1xPBS rinsed in ddH2O and air-dried. The slides were mounted in DABCO/20 mM Tris and imaged on a Zeiss Axioskop 2 at 100x magnification.

## Direct-repeat sister-chromatid recombination assay

Single colonies were grown in synthetic complete (SC) media lacking uracil with raffinose as a carbon source for 24 h at 30 °C. To determine spontaneous recombination rates, appropriate dilutions were plated on YPD media to determine viability, and on SC-LEU, and SC-URA-LEU to measure recombinants. 2% galactose was added to the cultures to induce expression of the HO endonuclease, which was expressed for 2 h before the cells were collected and plated to determine DSB-induced recombination frequencies. After 3 days of growth at 30 °C, colonies were counted from at least 11 independent cultures per genotype by the method of the median[100].

## D-loop capture assay (DLC) assay

For DLC experiments, all strains were in the W303 *RAD5* background. They contain a copy of the GAL1/10-driven HO endonuclease gene at the *TRP1* locus on chromosome IV (Chr. IV). A point mutation inactivates the HO cut site at the mating-type locus (MAT) on Chr. III (*MATa-inc*). The DSB-inducible construct contains the 117-bp HO cut site, a 2086-bp-long homology A sequence ( + 4 to + 2090 of the *LYS2* gene), and a 327-bp fragment of the PhiX174 genome flanked by multiple restriction sites. The DLC assay was conducted following established protocols with slight adjustments[101–103]. Specifically, zymolyase lysed cells were immediately subjected to restriction digestion, ligation, and DNA purification steps following hybridization with oligonucleotides[104]. The control experiments monitoring DSB formation, ligation efficiency, and a normalization locus are shown in Supplementary Fig. 4.

## Determination of BIR frequency

BIR frequency determination followed the method outlined in ref. 105. In brief, cells were cultivated to exponential phase in YEP-lactate medium and then plated on YPD plates. After 3 days, colonies were counted and subsequently replicated onto synthetic complete medium lacking lysine (*LYS2* drop- out) plates or YPD plates containing geneticin (G418). Cell viability post-HO induction was calculated by dividing the number of colonies on YP galactose plates by those on YP glucose plates. The percentage of cells repairing via BIR was determined by dividing the number of cells on LYS2 drop-out plates by the number on YP galactose plates, normalized to the number on YPD. BIR frequencies were determined three times for each strain.

## Detection of physical BIR products

Cells were cultured in YEP-lactate medium, followed by the addition of 2% galactose to induce HO endonuclease expression. Genomic DNA (25 ng) was then subjected to PCR amplification using Phusion High-Fidelity DNA Polymerase with the following cycling conditions: initial denaturation at 98 °C for 30 s, followed by 25 cycles of denaturation at 98 °C for 10 s, and annealing/extension at 72 °C for 150 s in a 25 μl reaction volume. BIR product detection utilized P1 and P2 primers, while HO cut detection employed D1 and D2 primers. Normalization of P1-P2 and D1-D2 products was achieved using C1 and C2 primers, as described previously[105,106].

## Analysis of sister chromatid junctions (SCJs)

Yeast cells were grown at 30 °C in a supplemented minimal medium (SMM). For G1 synchronization, cells were grown to mid-log phase and α-factor was added twice at 60 min intervals at 2 μg/ml (*BAR1* strains). Then, cells were washed three times and released into fresh medium with 50 μg/ml pronase in the absence or presence of MMS at the indicated concentrations. SCJs (X-shaped molecules) were analyzed by 2D-gel electrophoresis from cells arrested with sodium azide (0.1% final concentration) and cooled down on ice as reported[107]. Briefly, total DNA was isolated with the G2/CTAB protocol, digested with *Eco*RV and *Hind*III, resolved by neutral/neutral two-dimensional gel

electrophoresis, blotted to Hybond™-XL membranes, and analyzed by hybridization with the $^{32}$P-labeled A probe. Signals were quantified in a Fuji FLA5100 with the ImageGauge analysis program.

### In vivo ChEC analysis

Chromatin endogenous cleavage (ChEC) of *RAD51-MN*, *rad51-KN-MN* and *rad51-ED-MN* cells was performed as reported[35] from cultures grown at 30 °C in SMM with or without 0.05% MMS for 2 h and arrested with sodium azide (0.1% final concentration). For cleavage induction, digitonin-permeabilized cells were incubated with 2 mM CaCl$_2$ at 30 °C under gentle agitation. Total DNA was isolated and resolved into 0.8% TAE 1× agarose gels. Gels were scanned in a Fuji FLA5100, and the signal profile quantified using ImageGauge. The area of the DNA digestion profiles was equalized to eliminate DNA loading differences.

### Direct post-replication repair analysis in cells

All strains used for the iDamage assay are derivatives of the EMY74.7 strain[108]. In order to study tolerance events, all strains are deficient in repair mechanisms: nucleotide excision repair (*rad14*) and mismatch repair (*msh2*). Gene disruptions were achieved using PCR-mediated seamless gene deletion[109] or URAblaster[110] techniques. Rad51 mutations were introduced using CRISPR/Cas9[111]. Integration of plasmids carrying the (6–4)TT and N2dG-AAF lesions (or control plasmids without lesion) was performed as previously described[50]. Lesions are inserted either on the lagging or leading strand. The lesion is located in the lacZ gene, allowing to score TS events as blue colonies. The non-damaged strand contains a + 2 frameshift inactivating the lacZ gene, serving as a genetic marker for strand discrimination. Lesion tolerance rates were calculated as the relative integration efficiencies of damaged vs. non-damaged vectors normalized by the transformation efficiency of a control plasmid in the same experiment. DA events are calculated by subtracting TLS events from the total lesion tolerance events. All experiments were performed at least in triplicate. For each experiment, between 2000 and 4000 colonies were counted. Since we have not observed any difference in lesion tolerance pathways between lagging and leading strands, we plotted in the graphs average value of pooled data for leading and lagging strands. Graphs and statistical analysis were done using GraphPad Prism applying unpaired t-test. Bars represent the mean value ± sd.

### Protein purifications

Rad51 mutant and wild type proteins were purified to homogeneity as described[112,113]. The host *Saccharomyces cerevisiae* strain (WDHY1611; Supplementary Table 1) had a chromosomal deletion of the *RAD51* gene to avoid contaminating mutant protein preparation by wild type Rad51 protein. In brief, the mutant and wild type proteins (native, untagged) were purified using a four-column purification strategy including: Q-Sepharose, Cibacron Blue, Hydroxyapatite, and Mono Q. Rad54[114] and RPA[115] were purified as described. GST-Rad55-His$_6$-Rad57 were purified as described (Liu et al. 2011b) and dialyzed into storage buffer containing 20 mM Tris-HCl pH 7.5, 0.2 M NaCl, 0.1 mM EDTA, 1 mM DTT and 50% glycerol. Sgs1 was expressed in *Spodoptera frugiperda* (*Sf*9) cells using the pFB-MBP-Sgs1 vector and purified by amylose and NiNTA affinity purification as previously described[116]. The N-terminal MBP tag was removed after the amylose purification step using PreScission Protease. Yeast Exo1 was purified from *Sf*9 cells with pFB-Exo1-FLAG by FLAG affinity and HiTrap SP HP (Cytiva) ion exchange chromatography as described previously[117]. Dna2 was expressed in the *S. cerevisiae* strain WDH668 from the pGAL18 Flag-HA-Dna wt-His: vector and purified by NiNTA and FLAG affinity chromatography as described[118,119]. Yeast RPA was expressed using the p11d–rPA vector (a kind gift from M. Wold, University of Iowa) in BL21 (DE3) pLysS cells and purified on a HiTrap Blue column (Cytiva) followed by desalting with a HiTrap Desalting column (Cytiva) and finally with a HiTrap Q column (Cytiva)[120].

### Biochemical assays

ATPase assay: The ATP analysis was carried out using thin layer chromatography (TLC), as described[121]. 3 μM Rad51 was pre-incubated with 9 μM (nt or bp) DNA (phiX174 ss- or dsDNA), at 30 °C, in 20 mM Tris HCl (pH 7.5), 10 mM MgCl$_2$, 0.1 mg/ml BSA, 1 mM DTT, and 200 mM NaCl, then 0.05 mM ATP (1:100 nt radioactive) was added to initiate the reaction and start the time course. One μl aliquots were then spotted onto cellulose polyethyleneimine TLC plates. The plates were air dried and developed in 1 M formic acid and 0.5 M LiCl. The amount of ATP hydrolyzed was analyzed by PhosphorImager and quantified using ImageQuant software (Molecular Dynamics, Inc., Sunnyvale CA, USA).

DNA binding assays: DNA binding assays were performed as described[122] and conducted as salt titrations to assay the capability of Rad51 to bind DNA (ss- or dsDNA) in the presence of increasing concentrations of salt. DNA (10 μM nt/bp, ΦX174), salt (0–600 mM NaCl), and Rad51 (3.33 μM), were incubated for 15 min at 30 °C in the following buffer: 20 mM TEA, 1 mM DTT, 25 μg/ml BSA, 4 mM magnesium acetate, and 2.5 mM ATP. The reactions were stopped by the addition of glutaraldehyde (0.25%) and incubated for an additional 15 min at 30 °C, before being analyzed on 0.8% agarose gels in TAE buffer (40 mM Tris base, 20 mM acetic acid, 1 mM EDTA) 3.75 V/cm for 100 min. The dried gels were analyzed by PhospholImager and quantified using ImageQuant software (Molecular Dynamics, Inc., Sunnyvale CA, USA). The data were plotted using Prism GraphPad software which also determined the salt-titration midpoint. Additional DNA binding assays were conducted as protein titrations. DNA (3 μM nt/bp, olWDH2086 ssDNA or olWDH2086 annealed to olWDH2184 dsDNA) and Rad51 (250, 500, 1000, or 2000 nM) were incubated for 30 min at 30 °C in the following buffer: 35 mM Tris-acetate (pH 7.5), 50 or 100 mM NaCl, 7 mM magnesium acetate, 2 mM ATP, 1 mM DTT, and 0.25 mg/mL BSA. The reactions were analyzed on 1% 1x Tris-acetate agarose gels (Lonza) supplemented with 1 mM ATP and 2 mM magnesium acetate in TA buffer (40 mM Tris base, 20 mM acetic acid) at 3.33 V/cm for 60 min. Gels were imaged using a GE Amersham Imager 600. All bands were quantified through densitometry using Image Lab (Bio-Rad). Graphpad Prism was used for further analysis.

D-loop assay: Rad51-catalyzed D-loop assays were performed with slight modifications to the previously described methods[123]. A 5′ Cy5-labeled ssDNA oligonucleotide (25 nucleotides, olWDH2182 Supplementary Table 3) was annealed with another ssDNA oligonucleotide (100 nucleotides, olWDH2183 Supplementary Table 3) to serve as a tailed substrate. 0.25 μM wild-type Rad51, Rad51-ED, or Rad51-KN were incubated with 10 nM tailed substrate for 10 min at 30 °C in a buffer containing 35 mM Tris-acetate (pH 7.5), 50 or 100 mM NaCl, 7 mM magnesium acetate, 2 mM ATP, 1 mM DTT, 0.25 mg/mL BSA, 20 mM phosphocreatine, and 100 μg/mL phosphocreatine kinase. Next, 42 nM RPA was added to the reaction mixture for another 10-minute incubation. Finally, Rad54 (120 nM) and supercoiled plasmid dsDNA (10 nM) were added, with samples taken at 0, 5, 10, 20, and 30 min as indicated. For the experiments testing the Rad54 dependence (Supplementary Fig. 9D), 20 nM of 95-mer (olWDH566, Supplementary Table 3) and Rad51 (0.67 μM) were incubated for 10 min at 30 °C in buffer (30 mM TrisOAc pH 7.5, 1 mM DTT, 50 mg/ml BSA, 5 mM MgOAc, 20 mM Phosphocreatine, 4 mM ATP, and 0.1 mg/ml creatine kinase). RPA (0.1 μM) was added and incubated for another 10 min at 30 °C, followed by addition of Rad54 (0.112 μM) and dsDNA plasmid (20 nM molecules pUC19) to initiate the reaction and start the time course at 30 °C. Samples were then deproteinized, separated on 1% agarose gels, and documented using a GE Amersham Imager 600. All bands were quantified through densitometry using Image Lab (Bio-Rad).

Preparation of oligonucleotide-based substrates: The oligonucleotide-based DNA substrate used for the helicase and nuclease assays was generated by labeling the BIO100C oligonucleotide (see Supplementary Table 1 for oligonucleotides sequence) at the

3' end using terminal transferase (New England Biolabs) and [α-32P] dCTP (Hartmann Analytic) according to manufacturer's instructions and purified on G25 columns (Cytiva). The purified oligonucleotide was mixed with a two-fold excess of BIO100 oligonucleotide (see Supplementary Table 1 for oligonucleotides sequence) in annealing buffer (10 mM Tris-HCl pH 8, 50 mM NaCl, 10 mM MgCl$_2$), heated to 95 °C for 3 min and cooled down to room temperature overnight.

Helicase and nuclease assays: Helicase and nuclease assays were performed with a 100 bp substrate (1 nM, in molecules, oligonucleotides BIO100C and BIO100) in reaction buffer containing 25 mM Tris-acetate pH 7.5, 1 mM dithiothreitol, 2 mM magnesium acetate, 1 mM ATP, 80 U/ml pyruvate kinase (Sigma), 1 mM phosphoenolpyruvate and 0.15 mg/ml bovine serum albumin (New England Biolabs), in a final volume of 15 µl. The reactions were performed in the presence of 45 nM (final) *Saccharomyces cerevisiae* RPA. The indicated Rad51 variant was added to the reaction followed by the indicated resection factors, Sgs1 (100 nM) for the helicase assays, and Exo1 (25 nM) or Dna2-Sgs1 (1 nM each) for the nuclease assays. The reaction was incubated for 30 min at 30 °C and stopped by the addition of 5 µl of 2% STOP buffer (150 mM ethylenediaminetetraacetic acid [EDTA], 2% sodium dodecyl sulfate, 30% glycerol, bromophenol blue) supplemented with a 20-fold excess of unlabeled oligonucleotide with the same sequence as the labeled one (to prevent reannealing) and 1 µl of Proteinase K (18 mg/ml) (Roche). After deproteination for 30 min at 37 °C, the reactions were separated on a 10% TBE gel. The gel was dried on 17 Chr paper (Whatman), imaged on a Typhoon Imager (Cytiva) and quantitated using the ImageJ software.

Protein pulldown assay: 100 nM GST–Rad55–His$_6$–Rad57 or GST (Pierce) were incubated with 100 nM Rad51 in buffer P containing 25 mM Tris-acetate (pH 7.5), 10 mM magnesium acetate, 50 mM NaCl, 1 mM DTT, 10% glycerol and 0.01% NP-40 in a 25 µl volume for 1 h at room temperature. Equilibrated and BSA-treated glutathione agarose beads (Pierce) were added to the mixture and incubated for 1 h. The beads and supernatant were separated by centrifugation and the beads were washed twice with binding buffer P. The pulled-down protein complexes were eluted by heating at 65 °C for 10 min in 12 µl 2x SDS–PAGE loading buffer and separated through a 10% SDS–PAGE gel. Anti-Rad51 (1:1000, sc-133089, Santa Cruz Biotechnology,) and anti-GST (1:5000, 27457701, Cytiva) antibodies were used for immunoblotting. Protein bands were visualized using Clarity Western ECL substrate (Bio-Rad) on immunoblots and quantified using Image Lab (Bio-Rad). 5% of the input and 100% of the elution for each reaction were loaded for comparison.

## Electron microscopic imaging of Rad51 filaments

Rad51 filaments were assembled by incubating 2.5 µM Rad51, Rad51-ED, or Rad51-KN with 7.5 µM (bp or nt) of 1000 bp dsDNA or 700 nt ssDNA derived from phiX174 (*Ssp*I-*Ahd*1 fragment) at 30 °C for 10 min in a buffer containing 35 mM Tris-acetate (pH 7.5), 50 mM NaCl, 4 mM MgOAC, 2 mM ATP, and 1 mM DTT. The reaction mixture was then freshly diluted tenfold in 10 mM Tris-HCl (pH 7.5), 50 mM NaCl, and 5 mM MgCl$_2$. Samples were applied to 400-mesh carbon-coated copper grids (Ted Pella), negatively stained with 2% uranyl acetate, blotted dry, and then air-dried. Grids were imaged using a Talos Transmission Electron Microscope (Thermo Scientific), with images randomly captured from different grid regions. For each condition, 2–4 grids were used, and images were taken at 11,000x or 36,000x magnification. A total of 500 filaments per condition were manually measured using ImageJ.

## Statistical analysis

The 95% or better confidence intervals for the median were calculated using the information provided in Supplementary Table 4. This utilizes the formula: $(n+1)/2 \pm 1.96\,(\sqrt{n}/2)$ to provide the upper and lower limit of the 95% or better confidence interval from the rank order of values.

As such, this method will generate an uneven ± confidence interval around the median, making it easier to compare multiple bar graphs on a single figure. The p-values were always generated between two data sets using pairwise unpaired t-test with Welch's correction that does not assume equal standard deviations between data sets.

## Data availability

All data supporting the findings of this study are available within the paper and its Supplementary Information. Source data are provided with this paper.

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

## Acknowledgements

We thank Steve Kowalczykowski and Neil Hunter for sharing equipment, Rodney Rothstein and Lorraine Symington for strains, JoAnne Engebrecht for letting us use her microscope, and Akira Shinohara for anti-Rad51 serum. This work was supported by grants GM58015 and GM131037 from the US National Institutes of Health (WDH), Cancer Center Core Support Grant NCI P30CA093373 to UCD, grants BFU2015-63698-P and PID2021-127486NB-100 from MCIN/AEI/10.13039/501100011033 and "ERDF A way of making Europe" (F.P.), the Swiss National Science Foundation (SNSF) (Grants 310030_207588 and 310030_205199) and the European Research Council (ERC) (Grant 101018257) (P.C.), and Fondation pour la Recherche Médicale [FRM-EQU201903007797] (V.P.). D.M. was partially supported by the NCI T32 Oncogenic Signaling and Chromosome Biology training program T32 CA108459, S.J.C. was partially supported by the NIH T32 Molecular & Cellular Biology training program T32 GM007377; C.F. was partially supported by NIH postdoctoral fellowship F32GM083509; C.E. was supported by a postdoctoral fellowship from the Deutsche Forschungsgemeinschaft (DFG); MIC-L was supported by a fellowship from Spanish government FPU13/00955.

## Author contributions

D.M., S.K.G., J.L., S.J.C., S.H.H., G.R., M.I.C.L., K.H.M., F.V., C.F., and C.E. conducted the experiments; J.L. and S.G. conducted the structural visualization; D.M., F.P., P.C., V.P., and W.D.H. designed experiments and interpreted results; W.D.H. designed the study and wrote the manuscript with contributions from D.M., S.K.G., J.L., F.P., V.P., and P.C.

## Competing interests

The authors declare no competing interests.
