## [Transparent Peer Review file · Nature Communications]

Rad51 determines pathway usage in post-replication repair

Corresponding Author: Dr Wolf-Dietrich Heyer

Version 0:

Reviewer comments:

Reviewer #1

(Remarks to the Author)

In this manuscript, the authors report the identification of novel mutations in *Saccharomyces cerevisiae* Rad51 that maintain its homologous recombination (HR) function but impair fork protection. First, they found two Rad51 mutations (rad51-E135D (ED) and rad51-K305N (KN) that suppress the MMS-sensitivity of a rad51 Δ /rad54 Δ (or rad51 Δ /rad54ts) double mutant. The rad51-ED and -KN mutations induce an enhanced mutation rate and accentuate the MMS sensitivity of cells defective in post-replication repair, suggesting a shift in DNA repair pathway choice from HR to alternate pathways. Biochemical analyses with rad51-ED and rad51-KN proteins show that they have diminished activities in ATP hydrolysis, DNA binding, and DNA protection against nucleases, even though both appear to be proficient in D-loop formation. Additionally, the authors present predicted 3D-structure of the Rad51-ssDNA filament reconstituted with the mutants, and provides insight into the structural basis of the alteration of Rad51 interaction with duplex DNA.

The study is of general interests. However, I believe some claims could be more convincing if the authors would examine other rad51 mutants in parallel. Additionally, it is unclear whether the presented phenotypes (e.g., MMS sensitivity), interpreted as changes in pathway choice (HR, template switch, and translesion DNA synthesis), are solely due to Rad51's dsDNA binding or if they are outcome of some other effects of the mutations.

1. The rad51-ED and rad51-KN mutants were initially isolated from a screen using a double rad51/rad54 mutant (Fig. 1B), but subsequent studies with genomic integration of the mutations do not show similar suppression (Fig 1E). The authors mention this difference on Page 7 but the difference should be clarified further.
2. The mutator phenotype observed in rad51-ED and rad51-KN, despite their proficiency in HR, is an interesting finding. One wonders if this is specific to these two mutants or a general phenotype of rad51 mutants with decreased DNA binding affinity. The authors should test additional hypomorphic DNA binding mutations that are equivalent to, e.g. T131P or I131A in human RAD51. This would add significantly to the study regardless of the outcome of the suggested experiments.
3. Fig 4 & Fig S4: NaCl titration was used to examine DNA binding activity on ssDNA and dsDNA. However, protein titration that can be interpreted in terms of Kd would provide more informative insights than NaCl titration. Additionally, the use of long ssDNA may underestimate differences in ssDNA binding due to secondary structures forming in the substrate. I suggest testing shorter DNA substrates with a more quantitative method, e.g. MST.
4. Fig 7F: The D-loop assay was performed under low-salt condition, where the mutant does not show much difference in DNA binding affinity, as indicated in Fig 7C. It is important to standardize conditions for DNA binding and D-loop reactions to render conclusions robust.
5. Results from the D-loop assay (Fig. 7F) indicate that rad51-ED exhibits 9same can be said about rad51-KN, albeit to a lesser degree) more robust activity compared to wild type and the KN mutant proteins. The authors need to clarify whether this difference is due to intrinsic, enhanced recombinase activity of rad51-ED or resistance of D-loop coated by rad51-ED to Rad54-mediated D-loop dissociation.
6. Fig 8: The dsDNA binding properties and DNA protection observed are consistent with previous reports on human proteins, and the results align with expectation based on their DNA binding activity. It will be good to test additional other rad51 mutants (see point #4).

Reviewer #2

(Remarks to the Author)

Meyer and Ceballos et al. investigate a role of central HR recombinase Rad51 in post-replication repair using separation-of-

function mutants that maintain recombination activities but are specifically defective in binding to dsDNA with consequent changes to in vivo repair results. The multitude of genetic assays and in vitro characterization agree with the authors' model implicating a functional role of yeast Rad51 binding to dsDNA at stalled replication forks modulating pathway choice. The authors infer potential structural mechanisms for the two mutants characterized by comparing mutant AlphaFold predictions with an experimentally derived structure, but it is not clear the evidence provided is sufficient to justify the authors' assertions. Overall, the manuscript provides an interesting dissection of a non-standard role of Rad51 through convincing and logical experiments, but the structural evidence is currently too speculative and would need to be supported by additional experimental structures or other techniques.

Major comments:

1. The mutants were isolated in a gain-of-function genetic screen for alleles that allow MMS resistance in a temperature sensitive rad54 strain. It appears that this phenotype requires these alleles be on a high copy plasmid and no MMS resistance over wild-type protein was seen in native locus replacements, and additionally the MMS resistance with high copy plasmid is much weaker in a rad54 delete than rad54ts at non-permissive temperature. Presumably the high copy plasmid requirement means over-expression is necessary, but this is not specifically established. It is not clear how this screen would lead to the mutants and eventual phenotypes found. It would be helpful for the authors to discuss this and perhaps speculate on what is happening on a high copy plasmid with these mutants vs wild-type protein.
2. The authors did not write the method section of AlphaFold-prediction. Were ssDNA, ADP-AIF3, and magnesium included in the prediction to match the same cryoEM condition?
The authors compared individual side chain movements determined by cryoEM to the mutants that were predicted by AlphaFold. To validate prediction of side chain positions in AlphaFold, a very important thing to do first is the same AlphaFold prediction of WT-Rad51 and compared to cryoEM structure of WT-Rad51. Would side chain positions match each other? Would comparison of helices positions look different from the AlphaFold predictions of WT and the mutants like Fig 9C? if the authors can show WT-Rad51 AlphaFold prediction and is indeed look different from the mutants in overall filament shape + side chains, then I am convinced. However, if the WT-Rad51 AlphaFold prediction looks more similar to other mutants AlphaFold prediction, these differences the authors show in the paper may come from different methodology (cryoEM vs AlphaFold predictions).
3. The authors claim on page 16 to be "detecting the movement of amino acid side chains in the Rad51-ssDNA structure", but this seems to be more of an inference based on different protomer positions in the filament structure. This language seems to imply a more direct technique was used. Additionally, the expected change is on dsDNA while the experimental structure is on ssDNA with modeled dsDNA to infer mechanism. How do the authors know there aren't differences in the relevant side chain position/dynamics on ssDNA vs dsDNA? This is particularly relevant when very small changes are implied to cause an effect (see next comment). Assertions throughout, including the statement on page 17 in the first paragraph of the discussion "showing that the mutants render Rad51 less dynamic and more rigid" is not convincing from the modeling.
4. Pg16. "The Rad51-KN mutant has strengthened the interaction with M301 and A309, shortening the H-bond length from 3.1 Å to 3.0 Å, and 3.1 Å to 2.9 Å, respectively (Fig. 9D)"

I don't think this is valid argument...H-bond 3.1 to 3.0 to 2.9...

5. In this paper, the authors compared a cryo-EM structure of yeast Rad51-ssDNA filament with ADP-AIF3 (previously determined) with AlphaFold predicted structures of mutants (Rad51-ED and Rad51-KN). In the abstract and introduction parts, these are not described clearly in a way that readers may expect to see new CryoEM structures at 2.4Å resolution in the result section after reading the abstract.
6. Overall, the paper is too long with repetitive descriptions, may make readers lose interest while reading. Few descriptions/data need to be shortened.

Minor comments:

1. The last full sentence on page 3 is not clear/potential typo.
2. Typo on page 15 "Wild type Rad51 inhibited DNA degradation by Rad51"

Reviewer #3

(Remarks to the Author)
NCOMMS-24-45732-T

The MS by Meyer et al. describes the identification and characterization of rad51 mutants that are efficient in HR but defective in HR-mediated template switching. In contrast to the null mutant, the rad51-E135D (rad51-ED) and -K305N (rad51-KN) mutants are genetically efficient in HR, based on HO-triggered sister chromatid assays. The purified Rad51-ED and -KN are at least as efficient in D-loop formation as wild-type Rad51. In contrast, rad51-KN is partially sensitive to MMS and rad51-ED is as sensitive to MMS as the rad51 null mutant. These mutants show strong synergy with post-replicative repair mutants in MMS sensitivity, suggesting their defects in the generation of HR-mediated intermediates at replication forks, which is supported by their 2D gel analysis. They also showed that the post-replicative repair pathway is shifted to translesion synthesis in these rad51 mutants. Biochemically, these two mutant proteins, especially the rad51-ED, exhibit a reduction in ATPase activity and DNA binding to both single-stranded (ss) and double-stranded (ds) DNA, with dsDNA binding more severely affected. Consistently, these mutant proteins fail to protect dsDNA from degradation by exonucleases (Exo1 and Dna2-Sgs1) and the Sgs1 helicase. Taken together, it is proposed that Rad51 binding to duplex DNA is critical to

control the use of the pathway in post-replicative repair.

This MS would provide valuable information if these two rad51 mutations are indeed clear separation-of-function alleles. On the other hand, it is always a bit tricky to interpret the data obtained with hypomorphic mutants. Regarding their phenotypes in post-replicative repair, rad51-ED is similar to the rad51 null, while rad51-KN is partial. Regarding biochemistry, their failure to protect dsDNA from exonucleases and the Sgs1 helicase can be explained by their defect/reduction in dsDNA binding. The question is how good are they at HR, which is claimed to be as good as wild-type based primarily on HO-triggered sister chromatid assays and D-loop assays. I feel I need to be further convinced of their HR proficiency.

Please provide line numbers for ease of review.

(1) As explained above, rad51-ED is similar to rad51-null, while rad51-KN has partial MMS sensitivity, also in synergy with post-replicative repair mutants. In contrast, HR is fully competent in the sister chromatid assay. The proficiency of HR needs to be more concretely validated by following the mating type switching process initiated by the GAL-induced HO endonuclease by Southern blotting. In this way, not only the final efficiency of HR but also the kinetics can be clearly monitored.

(2) Regarding the D-loop assay shown in Fig. 7F. The data certainly show that the D-loop formation efficiency of rad51-ED and KN is at least as high as the wild-type level (or higher). On the other hand, Figure 7A-E clearly show that these proteins are impaired in ATP hydrolysis and DNA binding. Although ATP hydrolysis may not be very relevant to D-loop formation, ssDNA binding, as I understand, is a prerequisite for D-loop formation. Given that the physiological ionic strength is ~120 mM NaCl (page 14, 6th line from the bottom of the 1st paragraph), D-loop assay results within this range (i.e., 0~120 mM) should also be presented and their HR performance should be evaluated with these results included.

(3) The aim and rationale of the suppressor screen needs to be described in more detail. I found the relevance of using rad54 particularly confusing. I understand that Rad54 is not required for fork protection in mammals. The question is, is Rad54 required for fork protection in *S. cerevisiae*? MMS is introduced as a fork stalling agent, but according to Figure 1C, rad54 is as sensitive to MMS as rad51. Overall, I felt that the screening strategy was poorly explained, so it was hard to understand what the goal and principle of this genetic screen was.

(4) Related to (3), please explain how the identified properties of these mutants account for their suppression of rad54 MMS sensitivity.

(5) Fig. 6B, rad51-ED ChEC result is not convincing. The main band of the MMS-treated rad51-ED sample (15 min) shows a significant reduction in intensity. How could the band lose intensity without forming a smear?

Minor points

(6) Fig. 6A. Please include rad51-del (*sgs1-del*) in the 2D gel analysis, which is necessary to establish the baseline for assessing the phenotypes of rad51-KN and -ED.

(7) Fig. 4. Please be consistent with the MMS sensitivity assay. *rev3* and *pol30* are spot assays while *mms2* is a colony formation assay, making them difficult to compare. Please also include rad5-G535R results here. The results shown in Fig. S1AB are very confusing as the rad5-G535R results are included without any explanation until page 11. Also, please include the rad5-G535R single mutant in the spot assay, which is necessary to interpret the result.

(8) Fig. 3 title "rad51-ED and rad51-KN display a strong mutator phenotype similar to a complete HR defect". It seems that the mutator phenotype of rad51-ED when combined with rad54 is stronger than rad51-del (24x vs 10x), so the title is misleading. It would also be helpful to explain how this could happen.

(9) Fig. 5A. Overall, the system is poorly explained; everything is so small, and some critical information for understanding the system is missing. For example, what is *lox71*? Does it mean anything without explaining what *lox66* is? Which strand is in frame and which is not? etc. Please also clarify if rad51-ED and -KN are different from null. Also include how many colonies were examined per session.

(10) Page 20, last line of the 2nd paragraph "Alternatively, it is possible that ..."
It was not clear to me what exactly this line means.

(11) Fig. 1B. It was not clear which experiment (or both) was done at 37C (although I can certainly guess that ts experiments were done at 37C while the null was done at 30C).

Version 1:

Reviewer comments:

Reviewer #1

(Remarks to the Author)

The authors have done a very good job addressing comments raised. Specifically, they have provided new data (Figs. 4, 5, and S7) and addressed questions raised regarding the nature of the rad51-Il3A and S181P mutants in the Discussion.

This is a nice contribution and I do not have any more question.

Reviewer #3

(Remarks to the Author)

The manuscript by Meyer et al. originally described the identification and characterization of rad51 mutants that are efficient in homologous recombination (HR) but defective in post-replicative repair, namely rad51-E135D (rad51-ED) and rad51-K305N (rad51-KN). In response to reviewers' criticisms, the authors re-examined the impact of these mutations on HR and found the defects to be more extensive than initially reported. Specifically, both rad51-ED and rad51-KN exhibited severe HR defects when homologous sequences were located on different chromosomes. The authors accordingly stepped back from their original claim that HR can be separated from the activity required for post-replicative repair.

Overall, I have major concerns with aspects of the data interpretation:

(1) Fig. 4A

The authors claim that Rad51-ED binds ssDNA (100 nt) as efficiently as wild-type Rad51, and that Rad51-KN does so even more efficiently. In contrast, the data suggest the opposite: both mutant proteins show a marked reduction in ssDNA binding. This is evident when examining the shifted bands and intermediates represented by the smear. If only the unshifted band is considered, one could plot the graphs and obtain the reported Kd values. However, this approach does not accurately reflect the binding activity. The more appropriate conclusion from Fig. 4 is that both Rad51-ED and Rad51-KN are defective in ssDNA and dsDNA binding.

(2) Fig. S6

Consistent with Fig. 4, the results in Fig. S6 also indicate that Rad51-ED and Rad51-KN are defective in both ssDNA and dsDNA binding. Although the authors interpret the ssDNA defect as reflecting local duplex formation associated with a longer ssDNA substrate, the straightforward interpretation is that both mutant proteins are defective in ssDNA binding.

(3) Fig. 2B

The authors claim that spontaneous recombination, as assayed using a direct-repeat recombination system, is unaffected by rad51-ED and rad51-KN. However, the data do not convincingly support this conclusion. Spontaneous recombination appears to be mildly reduced in rad51-KN and more so in rad51-ED. The apparent lack of statistical significance may reflect an insufficient sample size, if a statistical test was performed at all.

The authors have already shown that ATP hydrolysis stimulated by ssDNA is reduced in Rad51-ED and Rad51-KN (Fig. 5A), more severely in Rad51-ED. Rad51 focus formation in response to MMS is also defective in both mutants (Fig. S8). These results are consistent with their defects in ssDNA binding.

In addition, Fig. 10A demonstrates that sister-chromatid joint formation is reduced in both rad51-ED and rad51-KN. Even in the sgs1 background, the data suggest that inter-sister association is compromised in these mutants, in agreement with my interpretation of Fig. 2B.

Taken together, the evidence strongly supports the view that these two mutants are hypomorphic alleles with defects across a broader spectrum of HR functions than proposed by the authors, with ED more severely affected than KN. Given that Rad51 has long been known to function in post-replicative repair, it is not surprising that partial defects in Rad51 activity produce corresponding partial phenotypes.

Furthermore, multiple studies have already reported that dsDNA binding of Rad51 is critical for protecting stalled replication forks from nucleases (refs. 30, 31). This further diminishes the novelty of the current work.

Minor points

(1) Line 219: there is no S2A (just S2).

(2) Fig. S2: The figure appears to show raw CFU counts rather than viability, despite being titled as viability. Additionally, the elevated levels of rad51-KN under both induced and spontaneous conditions are confusing and should be more clearly explained in the main text.

(3) Line 292: It would be useful to include ssDNA as well in the assay.

(4) Line 424: Should this refer to Fig. S3B rather than S3A?

(5) Line 425: The nature of the discrepancy mentioned is unclear and should be clarified.

(6) Line 493: I was puzzled to find that the FigS11 does not show SCJ formation in the rad51 null mutant. Regarding the importance of including the rad51 null mutant in Fig10A. No matter how many times it has been done previously, including a negative control is essential because it sets the base line for the data acquisition and interpretation.

(7) Fig. 10B: The phenotypes of strains expressing Rad51-KN-MN and Rad51-ED-MN should be compared directly with their untagged versions. The MN tag may alter the properties of these alleles.

Reviewer #4

(Remarks to the Author)

I have reviewed the responses to each critique made by all three reviewers. I think that by changing the terminology from separation of function mutants was appropriate, they addressed the biochemical concerns adequately by performing their binding assays with new substrates and their D-loop assays at higher salt concentrations, and by removing their alpha fold model structural analysis the authors have more than adequately addressed all reviewers concerns-particularly Reviewer 2.

We greatly appreciate the helpful and constructive comments by all reviewers. Below we discuss point-by-point our responses and how we modified the manuscript to implement the critiques. The text has been substantially revised and significant new data (Figures 3, 4, 5C-E, Figure S3C, S4, S5B,C, S7, S8, S9A-C, S11) were added, improving the manuscript significantly and offering additional insights into the mutants.

In short, while the mutants are fully proficient in inter-sister recombination, the analysis revealed a defect in inter-homolog recombination (new Figure 3, S4), which in contrast to sister chromatid recombination necessitates a genome-wide homology search. Moreover, additional biochemical analysis revealed that the mutants form shorter filaments (new Figures S7, S8), which is likely the reason for the inter-homolog HR defect.

The expanded biochemical analysis reinforced that the major defect of the mutants is binding to dsDNA, with good correlation between the *in vitro* and *in vivo* phenotypes showing a greater impact of the E135D mutation (new Figure 4, 5C-E, S7, S9A-C).

In conclusion, we step back from the claim of separation-of-function mutants. The mutants reveal differences between sister-chromatid and interhomolog recombination, that align well with recent results from the Piazza and Symington labs that genome-wide homology search differs from sister chromatid recombination. Importantly, our major conclusion stands reinforced that the mutants primarily affect dsDNA binding and that the mutant phenotypes *in vitro* and *in vivo* correlate in severity with defects in fork protection leading to a shift in pathway usage at stalled forks and on over-reliance on TLS and fork regression in the mutants. We have revised the discussion accordingly.

We added line numbers to the version with tracked changes (in red) to aid the reviewers.

Reviewer #1 (Remarks to the Author):

In this manuscript, the authors report the identification of novel mutations in *Saccharomyces cerevisiae* Rad51 that maintain its homologous recombination (HR) function but impair fork protection. First, they found two Rad51 mutations (rad51-E135D (ED) and rad51-K305N (KN)) that suppress the MMS-sensitivity of a rad51 Δ /rad54 Δ (or rad51 Δ /rad54ts) double mutant. The rad51-ED and -KN mutations induce an enhanced mutation rate and accentuate the MMS sensitivity of cells defective in post-replication repair, suggesting a shift in DNA repair pathway choice from HR to alternate pathways. Biochemical analyses with rad51-ED and rad51-KN proteins show that they have diminished activities in ATP hydrolysis, DNA binding, and DNA protection against nucleases, even though both appear to be proficient in D-loop formation. Additionally, the authors present predicted 3D-structure of the Rad51-ssDNA filament reconstituted with the mutants, and provides insight into the structural basis of the alteration of Rad51 interaction with duplex DNA.

The study is of general interests. However, I believe some claims could be more convincing if the authors would examine other rad51 mutants in parallel. Additionally, it is unclear whether the presented phenotypes (e.g., MMS sensitivity), interpreted as changes in pathway choice (HR, template switch, and translesion DNA synthesis), are solely due to Rad51's dsDNA binding or if they are outcome of some other effects of the mutations.

Thank you for recognizing the general interest of the results. We would not venture to say that the phenotype is 'solely' the consequence of the dsDNA binding defect, but that this defect best correlates with the phenotypic differences of the mutants. We made this point clearer in the text

(lines 538-41, 554, 637).

1. The *rad51*-ED and *rad51*-KN mutants were initially isolated from a screen using a double *rad51/rad54* mutant (Fig. 1B), but subsequent studies with genomic integration of the mutations do not show similar suppression (Fig 1E). The authors mention this difference on Page 7 but the difference should be clarified further.

The reviewer is correct in pointing out that the *rad51* mutants show significantly less suppression on plates in single copy (Figure S1A) than when overexpressed (Figure 1B), as was the case during the screen and the validation. Using a colony formation assay (Figure 1E) the genomic integration showed no suppression. We explain this difference between the results from Figure 1B and Figure S1A to be a consequence of overexpressing the Rad51 mutants during the screen and validation versus the genomic integration. The difference between plate assays which use serial dilutions (Figure 1B, S1B) and the colony formation assay in Figure 1E is explained by the former measuring a mixture of growth and survival, whereas the latter measures colony formation. We have added text to discuss these differences (lines 183-187).

2. The mutator phenotype observed in *rad51*-ED and *rad51*-KN, despite the ir proficiency in HR, is an interesting finding. One wonder if this is specific to these two mutants or a general phenotype of *rad51* mutants with decreased DNA binding affinity. The authors should test additional hypomorphic DNA binding mutations that are equivalent to, e.g. T131P or I131A in human RAD51. This would add significantly to the study regardless of the outcome of the suggested experiments.

It is unclear whether this is a general phenotype of hypomorphic *rad51* mutants. Human *RAD51-T131P* is a semi-dominant mutation, whose viability depends on the presence of wild type human RAD51 protein (PMID: 26253028). Our mutations were analyzed in a haploid context, which is not comparable. We are not familiar with an I131A mutation in human RAD51, and PubMed and Google searches were negative. The reviewer might be referring to the yeast *rad51-I13A* reported in Cloud et al. 2012 Science. This mutation is proficient in meiotic recombination providing a short filament seed for the Dmc1 filament but is deficient in the mitotic HR function leading to IR sensitivity like the *rad51* deletion mutant (Cloud et al. 2012). This phenotype clearly differs from our mutants which are not IR sensitive (*rad51*-KN) or significantly less than the deletion (*rad51*-ED). We also discuss a comparison with a recently published separation of function mutant in human RAD51, RAD51S181P. We added a brief discussion of these differences to the discussion on lines 609-622.

3. Fig 4 & Fig S4: NaCl titration was used to examine DNA binding activity on ssDNA and dsDNA. However, protein titration that can be interpreted in terms of Kd would provide more informative insights than NaCl titration. Additionally, the use of long ssDNA may underestimate differences in ssDNA binding due to secondary structures forming in the substrate. I suggest testing shorter DNA substrates with a more quantitative method, e.g. MST.

The reviewer has three concerns. First, the EMSA conducted as salt midpoint titration to measure relative differences in DNA binding affinity. EMSA and salt midpoint titrations are a well-established and accepted approach to compare DNA binding affinities, going back to the pioneering work of Peter von Hippel and being frequently used for recombination proteins by Steve Kowalczykowski and others. Second, the reviewer prefers a protein titration to allow determining apparent Kd. Third, the reviewer is worried that the ssDNA substrate used in our EMSA experiments may form secondary structures, which may lead to an underestimation of the ssDNA binding defect of the mutants. We accept the last two points and have conducted

protein titrations with 100 nt and 100 bp substrates to derive K_d values at 50 mM and 100 mM NaCl (new Figure 4), the same concentrations at which we conducted the new D-loop experiments (see point #4, new Figure 5C-E). As implied by the reviewer, the ssDNA binding defect using the longer substrate was likely related to its secondary structure, as both mutant proteins showed no binding defect to the 100 nt substrate. These results reinforce the conclusions that the mutant proteins affect dsDNA binding, and that Rad51-ED displays the larger effect than Rad51-KN.

We examined filament length by EM to further examine the dsDNA binding defect and found that both mutant proteins formed significantly shorter filaments (new Figure S7). We also examined filament formation *in vivo* by immunofluorescence (IF) and found that both mutants caused a strong defect in Rad51 filament formation in response to DNA damage (new Figure S8). This result suggests that also filament formation on ssDNA is affected and may provide an explanation for the defect in interhomolog recombination. In this respect our mutants are similar to the *rad51-II3A* mutant described by Dr. Bishop (see point #3), which also forms shorter filament that are undetectable by IF. This point was added to the discussion (lines 609-610).

These results were unexpected but provide significant insights into the mutants. We thank the reviewer for the helpful comments that made us reexamine this aspect.

4. Fig 7F: The D-loop assay was performed under low-salt condition, where the mutant does not show much difference in DNA binding affinity, as indicated in Fig 7C. It is important to standardize conditions for DNA binding and D-loop reactions to render conclusions robust.

We agree with the reviewer's point and added new D-loop experiments that were conducted at 50 mM and 100 mM NaCl using the same 100mer used for the new EMSA experiments (see response to point #3). These new data are presented in the new Figure 5.

5. Results from the D-loop assay (Fig. 7F) indicate that rad51-ED exhibits 9same can be said about rad51-KN, albeit to a lesser degree) more robust activity compared to wild type and the KN mutant proteins. The authors need to clarify whether this difference is due to intrinsic, enhanced recombinase activity of rad51-ED or resistance of D-loop coated by rad51-ED to Rad54-mediated D-loop dissociation.

The reviewer points out that in D-loop experiments using short invading ssDNA two phases exist, an early D-loop formation phase and a later D-loop dissociation phase. The D-loop dissociation is a function of Rad54, as we have shown in Wright and Heyer (2014) PMID 24486020. The reviewer makes a good point that the Rad51 mutant could affect either phase of this process. We conducted new D-loop experiments under the same salt conditions as the EMSA experiments using a tailed substrate (new Figure 5). The kinetics shows no difference in the formation phase, but faster dissociation of D-loops in the reaction with the mutant Rad51 proteins. This is consistent with a functional defect in binding dsDNA, as after D-formation Rad51 is bound to dsDNA, from where it needs to be displaced for D-reversal by Rad54 (Wright and Heyer (2014)).

6. Fig 8: The dsDNA binding properties and DNA protection observed are consistent with previous reports on human proteins, and the results align with expectation based on their NA binding activity. It will be good to test additional other rad51 mutants (see point #4).

It is unclear what can be gained by analyzing additional mutants, in particular the ones mentioned by the reviewer, see response to point #2. (The reviewer refers to #4, which appears

to be incorrect.)

Reviewer #2 (Remarks to the Author):

Meyer and Ceballos et al. investigate a role of central HR recombinase Rad51 in post-replication repair using separation-of-function mutants that maintain recombination activities but are specifically defective in binding to dsDNA with consequent changes to *in vivo* repair results. The multitude of genetic assays and *in vitro* characterization agree with the authors' model implicating a functional role of yeast Rad51 binding to dsDNA at stalled replication forks modulating pathway choice. The authors infer potential structural mechanisms for the two mutants characterized by comparing mutant AlphaFold predictions with an experimentally derived structure, but it is not clear the evidence provided is sufficient to justify the authors' assertions. Overall, the manuscript provides an interesting dissection of a non-standard role of Rad51 through convincing and logical experiments, but the structural evidence is currently too speculative and would need to be supported by additional experimental structures or other techniques.

We thank the reviewer for the supportive comments and agree that the structural modeling does not provide a satisfactory explanation for the biochemical and *in vivo* behavior of the proteins. Since we just had completed the high-resolution cryoEM structure of the yeast Rad51-ssDNA filament with ADP-AIF₃, which is now published in *Nucleic Acids Research* (Liu et al. 2025), we felt compelled to try to model the mutants. However, we realize now this analysis does not add to the manuscript and we eliminated this part. We only retained as supplemental Figure S5A-C the information on the sequence conservation around both residues and their locations with respect to bound DNA and nucleotide co-factor.

Major comments:3

1. The mutants were isolated in a gain-of-function genetic screen for alleles that allow MMS resistance in a temperature sensitive *rad54* strain. It appears that this phenotype requires these alleles be on a high copy plasmid and no MMS resistance over wild-type protein was seen in native locus replacements, and additionally the MMS resistance with high copy plasmid is much weaker in a *rad54* delete than *rad54ts* at non-permissive temperature. Presumably the high copy plasmid requirement means over-expression is necessary, but this is not specifically established. It is not clear how this screen would lead to the mutants and eventual phenotypes found. It would be helpful for the authors to discuss this and perhaps speculate on what is happening on a high copy plasmid with these mutants vs wild-type protein.

See above.

2. The authors did not write the method section of AlphaFold-prediction. Were ssDNA, ADP-AIF₃, and magnesium included in the prediction to match the same cryoEM condition? The authors compared individual side chain movements determined by cryoEM to the mutants that were predicted by AlphaFold. To validate prediction of side chain positions in AlphaFold, a very important thing to do first is the same AlphaFold prediction of WT-Rad51 and compared to cryoEM structure of WT-Rad51. Would side chain positions match each other? Would comparison of α helices positions look different from the AlphaFold predictions of WT and the mutants like Fig 9C? if the authors can show WT-Rad51 AlphaFold prediction and is indeed look different from the mutants in overall filament shape + side chains, then I am convinced. However, if the WT-Rad51 AlphaFold prediction looks more similar to other mutants AlphaFold prediction, these differences the authors show in the paper may come from different methodology (cryoEM vs AlphaFold predictions).

See above.

3. The authors claim on page 16 to be “detecting the movement of amino acid side chains in the Rad51-ssDNA structure”, but this seems to be more of an inference based on different protomer positions in the filament structure. This language seems to imply a more direct technique was used. Additionally, the expected change is on dsDNA while the experimental structure is on ssDNA with modeled dsDNA to infer mechanism. How do the authors know there aren't differences in the relevant side chain position/dynamics on ssDNA vs dsDNA? This is particularly relevant when very small changes are implied to cause an effect (see next comment). Assertions throughout, including the statement on page 17 in the first paragraph of the discussion “showing that the mutants render Rad51 less dynamic and more rigid” is not convincing from the modeling.

See above.

4. Pg16. “The Rad51-KN mutant has strengthened the interaction with M301 and A309, shortening the H-bond length from 3.1 Å to 3.0 Å, and 3.1 Å to 2.9 Å, respectively (Fig. 9D)”

I don't think this is valid argument...H-bond 3.1 to 3.0 to 2.9...

See above.

5. In this paper, the authors compared a cryo-EM structure of yeast Rad51-ssDNA filament with ADP-AIF3 (previously determined) with AlphaFold predicted structures of mutants (Rad51-ED and Rad51-KN). In the abstract and introduction parts, these are not described clearly in a way that readers may expect to see new CryoEM structures at 2.4Å resolution in the result section after reading the abstract.

See above.

6. Overall, the paper is too long with repetitive descriptions, may make readers lose interest while reading. Few descriptions/ data need to be shortened.

We shortened the descriptions and eliminated repetitions. Thank you for the comment.

Minor comments:

1. The last full sentence on page 3 is not clear/potential typo.
2. Typo on page 15 “Wild type Rad51 inhibited DNA degradation by Rad51”

Thank you for spotting these issues, which are now corrected. Lines 86-88, line 340.

Reviewer #3 (Remarks to the Author):

The MS by Meyer et al. describes the identification and characterization of rad51 mutants that are efficient in HR but defective in HR-mediated template switching. In contrast to the null mutant, the rad51-E135D (rad51-ED) and -K305N (rad51-KN) mutants are genetically efficient in HR, based on HO-triggered sister chromatid assays. The purified Rad51-ED and -KN are at least as efficient in D-loop formation as wild-type Rad51. In contrast, rad51-KN is partially sensitive to MMS and rad51-ED is as sensitive to MMS as the rad51 null mutant. These mutants

show strong synergy with post-replicative repair mutants in MMS sensitivity, suggesting their defects in the generation of HR-mediated intermediates at replication forks, which is supported by their 2D gel analysis. They also showed that the post-replicative repair pathway is shifted to translesion synthesis in these rad51 mutants. Biochemically, these two mutant proteins, especially the rad51-ED, exhibit a reduction in ATPase activity and DNA binding to both single-stranded (ss) and double-stranded (ds) DNA, with dsDNA binding more severely affected. Consistently, these mutant proteins fail to protect dsDNA from degradation by exonucleases (Exo1 and Dna2-Sgs1) and the Sgs1 helicase. Taken together, it is proposed that Rad51 binding to duplex DNA is critical to control the use of the pathway in post-replicative repair.

This MS would provide valuable information if these two rad51 mutations are indeed clear separation-of-function alleles. On the other hand, it is always a bit tricky to interpret the data obtained with hypomorphic mutants. Regarding their phenotypes in post-replicative repair, rad51-ED is similar to the rad51 null, while rad51-KN is partial. Regarding biochemistry, their failure to protect dsDNA from exonucleases and the Sgs1 helicase can be explained by their defect/reduction in dsDNA binding. The question is how good are they at HR, which is claimed to be as good as wild-type based primarily on HO-triggered sister chromatid assays and D-loop assays. I feel I need to be further convinced of their HR proficiency.

We thank the reviewer for the helpful comments that led to significant new experimentation which greatly improved our understanding of the functional defects of the Rad51 mutants. This analysis revealed a defect in inter-homolog recombination (new Figure 3), which in contrast to sister chromatid recombination necessitates a genome-wide homology search. Moreover, additional biochemical analysis revealed that the mutant form shorter filaments (new Figures S7, S8), which is likely the reason for the inter-homolog HR defect.

The expanded biochemical analysis reinforced that the major defect of the mutants is in binding to dsDNA, with good correlation between the *in vitro* and *in vivo* phenotypes showing a greater impact of the E135D mutation (new Figure 4).

In conclusion, we step back from the claim of separation-of-function mutants. The mutants reveal differences between sister-chromatid and interhomolog recombination, that align well with recent results from the Piazza and Symington labs that genome-wide homology search differs from sister chromatid recombination. Our major conclusion stands reinforced that the mutants primarily affect dsDNA binding and that the mutant phenotypes *in vitro* and *in vivo* correlate in severity with defects in fork protection leading to a shift in pathway usage at stalled forks and on over-reliance on TLS and fork regression in the mutants. We made numerous text changes accordingly.

Please provide line numbers for ease of review.

We added line numbers to indicate the changes in the version of the revised manuscript with tracked changes.

(1) As explained above, rad51-ED is similar to rad51-null, while rad51-KN has partial MMS sensitivity, also in synergy with post-replicative repair mutants. In contrast, HR is fully competent in the sister chromatid assay. The proficiency of HR needs to be more concretely validated by following the mating type switching process initiated by the GAL-induced HO endonuclease by Southern blotting. In this way, not only the final efficiency of HR but also the kinetics can be clearly monitored.

We agree with the reviewer and added two key experiments: First, we use our newly developed D-loop capture assay to directly track D-loop levels. Second, we used a BIR assay that allows to physically and genetically track the endpoints. These data are shown in Figure 3. Surprisingly and unexpectedly, both mutants showed a strong defect in interhomolog recombination that required a genome-wide homology search. Interestingly, this phenotype correlates with shorter RAD51 filaments, which may point to a difference between inter-sister and inter-homolog recombination in the requirement for long Rad51 filaments (Figures S7, S8).

(2) Regarding the D-loop assay shown in Fig. 7F. The data certainly show that the D-loop formation efficiency of *rad51-ED* and *KN* is at least as high as the wild-type level (or higher). On the other hand, Figure 7A-E clearly show that these proteins are impaired in ATP hydrolysis and DNA binding. Although ATP hydrolysis may not be very relevant to D-loop formation, ssDNA binding, as I understand, is a prerequisite for D-loop formation. Given that the physiological ionic strength is ~120 mM NaCl (page 14, 6th line from the bottom of the 1st paragraph), D-loop assay results within this range (i.e., 0~120 mM) should also be presented and their HR performance should be evaluated with these results included.

We agree with the reviewer and added new D-loop experiments under the same salt conditions as the new EMSA experiments at 50 mM and 100 mM NaCl. In fact, using a 100mer that is devoid of dsDNA potential revealed that the mutant proteins had no defect binding ssDNA in the EMSA assay. See also our responses to points #4, 5 of reviewer 1 on this topic.

(3) The aim and rationale of the suppressor screen needs to be described in more detail. I found the relevance of using *rad54* particularly confusing. I understand that Rad54 is not required for fork protection in mammals. The question is, is Rad54 required for fork protection in *S. cerevisiae*? MMS is introduced as a fork stalling agent, but according to Figure 1C, *rad54* is as sensitive to MMS as *rad51*. Overall, I felt that the screening strategy was poorly explained, so it was hard to understand what the goal and principle of this genetic screen was.

The reviewer is correct in noting that the MMS sensitivity of *rad51* and *rad54* cells is very similar but that does not necessarily imply identical mechanisms. We have improved Figure 1 and added clarification to the legend. The goal was to isolate novel *rad51* mutants that separated Rad51 function from Rad54 function. Reviewer 1 brought up a similar point (#1).

(4) Related to (3), please explain how the identified properties of these mutants account for their suppression of *rad54* MMS sensitivity.

Reviewer 1 brought up the same point (#1), and we addressed this there.

(5) Fig. 6B, *rad51-ED* ChEC result is not convincing. The main band of the MMS-treated *rad51-ED* sample (15 min) shows a significant reduction in intensity. How could the band lose intensity without forming a smear?

Even though we try to load similar amounts of total DNA, there are some differences as with the indicated case where the lower band intensity reflects less total DNA. To eliminate DNA loading differences, the area of the DNA digestion profiles was equalized as indicated in the methods section. We added some explanation to the text (line 781-782) and added a figure of the repeat experiment as Figure S11 as well as a comment that the results are reproducible.

Minor points

(6) Fig. 6A. Please include *rad51-del* (*sgs1-del*) in the 2D gel analysis, which is necessary to

establish the baseline for assessing the phenotypes of *rad51-KN* and *-ED*.

Rad51 is required for the formation of SCJs as determined by 2D gel analysis (less of 5% of the wild type levels are left in the *rad51* null mutant). This observation was first reported by Foiani and colleagues (Liberi et al. (2005) *Genes Dev.* 19:339; PMID 15687257). This has been reproduced several times; lastly by Branzei and colleagues (Joseph et al. 2022; *Nature Comm.* 13:2480; PMID 35513396 Figure 6B), using the same protocol and experimental conditions that we have used in the current study (replication and recombination intermediates from ARS305 in cells released from G1 in 0.033% MMS). As can be concluded from our analysis in Figure 6A, the *rad51-ED* mutant displayed a defect almost as severe as *rad51Δ* (10-15% of the wild type), whereas the effect of *rad51-KN* was much milder. We have added this information to the results section (lines 491-493).

(7) Fig. 4. Please be consistent with the MMS sensitivity assay. *rev3* and *pol30* are spot assays while *mms2* is a colony formation assay, making them difficult to compare. Please also include *rad5-G535R* results here. The results shown in Fig. S1AB are very confusing as the *rad5-G535R* results are included without any explanation until page 11. Also, please include the *rad5-G535R* single mutant in the spot assay, which is necessary to interpret the result.

The experiments in Figure 4A, B, F, G (now Figure 7) used serial dilution assays and C, D, E quantitative survival assays, because the *mms2* analysis in C-E involved far fewer strains. This more quantitative analysis with *mms2* allowed more detailed insights into the relationship between the *rad51* mutants and PCNA polyubiquitylation. The results clearly show the synergy/additivity between the *rad51* mutants and all PRR mutants in A-G. We are not sure, what the effort to conduct quantitative survival assays for another 48 strains would add.

Figure S3 (formerly Figure S1) serves two purposes, first to show the IR sensitivity of the mutants, and second, the show the effect of the hypomorphic *rad5-G535R* mutation. We have reorganized the figure to reflect this sequence and guide the reader through the figure to focus first on the IR data in *RAD5* wild type strains and then later on the *rad5-G535R* data.

We agree with the reviewer that the single *rad5-G535R* strain is missing from the analysis in Figure S1 (now Figure S3), as it was described and published to have no effect on growth and survival at his low MMS concentration. We added the direct comparison of *rad5-G535R*, wild type, and *rad51Δ* to Figure S3 as part C. We can no longer conduct the IR experiment in Figure S1B, as our cesium source has been decommissioned. *rad5-G535R* was described as IR-resistant, and we added this information to the text (line 430-432).

(8) Fig. 3 title "*rad51-ED* and *rad51-KN* display a strong mutator phenotype similar to a complete HR defect". It seems that the mutator phenotype of *rad51-ED* when combined with *rad54* is stronger than *rad51-del* (24x vs 10x), so the title is misleading. It would also be helpful to explain how this could happen.

We agree with the reviewer that the title does not reflect that particular result. We rephrased the title (now Figure 8) and added some speculation, as to how this additive effect could be explained on line 438, 450-451.

(9) Fig. 5A. Overall, the system is poorly explained; everything is so small, and some critical information for understanding the system is missing. For example, what is *lox71*? Does it mean anything without explaining what *lox66* is? Which strand is in frame and which is not? etc.

Please also clarify if *rad51-ED* and *-KN* are different from null. Also include how many colonies were examined per session.

The experimental system is now better explained in the text (lines 459-461). Specifically, the description of the use of *lox71* and *lox66* sites has been added to the figure legend and in the results section.

The figure has been improved according to the reviewer's suggestions. The text in the figure has been enlarged.

In response to the reviewer's questions about the strands: The strand in frame is the strand where the lesion has been inserted. Our system allows insertion either on the lagging or leading strand. Since we did not observe any difference, the data shown is an average of leading and lagging strand integration. This is now better explained in the method section (lines 790-793, 797-800). The number of colonies is also now mentioned in the method section.

As requested, we added the statistical tests comparing *rad51-ED* and *rad51-KN* to the *rad51* null mutant. Some of the differences are not significant, and some have a high p-value indicating low significance. Given the small difference and high p-value, the differences are unlikely to be biologically relevant.

Thank you for your comments to improve the presentation.

(10) Page 20, last line of the 2nd paragraph "Alternatively, it is possible that ..."
It was not clear to me what exactly this line means.

We have eliminated that discussion.

(11) Fig. 1B. It was not clear which experiment (or both) was done at 37C (although I can certainly guess that ts experiments were done at 37C while the null was done at 30C).

We apologize for the oversight, both experiments were conducted at 37C, and this is now noted in the figure legend in addition to notation in the figure.

Response to reviews

We thank reviewers 1 and 4 for accepting our revision and reviewer 3 for thoughtful comments that we are addressing in this second revision. These comments helped us to more accurately describe the results, and we appreciate the careful editorial comments.

Reviewer #1 (Remarks to the Author):

The authors have done a very good job addressing comments raised. Specifically, they have provided new data (Figs. 4, 5, and S7) and addressed questions raised regarding the nature of the rad51-II3A and S181P mutants in the Discussion.

This is a nice contribution and I do not have any more question.

Response:

Thank you for your supportive comments.

Reviewer #3 (Remarks to the Author):

The manuscript by Meyer et al. originally described the identification and characterization of rad51 mutants that are efficient in homologous recombination (HR) but defective in post-replicative repair, namely rad51-E135D (rad51-ED) and rad51-K305N (rad51-KN). In response to reviewers' criticisms, the authors re-examined the impact of these mutations on HR and found the defects to be more extensive than initially reported. Specifically, both rad51-ED and rad51-KN exhibited severe HR defects when homologous sequences were located on different chromosomes. The authors accordingly stepped back from their original claim that HR can be separated from the activity required for post-replicative repair.

Overall, I have major concerns with aspects of the data interpretation:

Response

We agree with major points 1 and 2 that there is a minor ssDNA binding defect that is not captured in the Kd and STM data but is indicated by the increased smear in the EMSA with ssDNA. This may be related to a subtle stability defect in ssDNA binding, which we now acknowledge in the discussion. This should not distract from the major difference between the strong dsDNA binding defect and the subtle ssDNA binding shown by three independent experiments (Figure 4, S6, S7).

(1) Fig. 4A

The authors claim that Rad51-ED binds ssDNA (100 nt) as efficiently as wild-type Rad51, and that Rad51-KN does so even more efficiently. In contrast, the data suggest the opposite: both mutant proteins show a marked reduction in ssDNA binding. This is evident when examining the shifted bands and intermediates represented by the smear. If only the unshifted band is considered, one could plot the graphs and obtain the reported Kd values. However, this approach does not accurately reflect the binding activity. The more appropriate conclusion from Fig. 4 is that both Rad51-ED and Rad51-KN are defective in ssDNA and dsDNA binding.

Response:

The reviewer is correct in pointing out that the data are plotted as bound DNA, which is the inverse of the unshifted substrate band. Our interpretation focused on the massive difference between ssDNA and dsDNA, which is reflected in the Kd data. The EMSA reveals additional information on the electrophoretic profile of the protein:DNA complexes. The reviewer points out that there is more smear of not fully shifted substrate in the reactions with mutants. We now acknowledge this in the discussion of the data on page 9 and reworded the section heading on page 8 accordingly. This is likely related to subtly altered stability in ssDNA binding.

(2) Fig. S6

Consistent with Fig. 4, the results in Fig. S6 also indicate that Rad51-ED and Rad51-KN are defective in both ssDNA and dsDNA binding. Although the authors interpret the ssDNA defect as reflecting local duplex formation associated with a longer ssDNA substrate, the straightforward interpretation is that both mutant proteins are defective in ssDNA binding.

Response:

As the reviewer pointed out in the previous round of review, the long DNA substrates used in Figure S6 form secondary structures with dsDNA hairpin stems. For this reason, we added the experiment in Figure 4 using ss oligonucleotide substrates devoid of potential for dsDNA formation to address this point. The EMSA data were quantified as in Figure 4 showing bound DNA, the inverse of free substrate. The salt titration midpoint (STM) is significantly more affected with dsDNA than with ssDNA. The reviewer is correct that the STM reduction for ssDNA reflects that a portion of the substrate was dsDNA, as was pointed out in the original review, and possibly altered affinity/stability in binding, as pointed out now. We acknowledge this latter possibility now specifically on page 9. Also, in Figure S6, the data clearly show a much stronger defect in dsDNA binding, for example at 0 mM NaCl there is no defect for ssDNA but a 20% or more reduction in binding dsDNA. Likewise, the STM values are much more reduced for dsDNA (58-75%), than for ssDNA (42-47%). Also the EM analysis of filaments (Figure S7) shows a stronger defect in binding dsDNA than ssDNA. These data independently validate the conclusion that dsDNA binding is much more affected than ssDNA binding.

(3) Fig. 2B

The authors claim that spontaneous recombination, as assayed using a direct-repeat recombination system, is unaffected by *rad51-ED* and *rad51-KN*. However, the data do not convincingly support this conclusion. Spontaneous recombination appears to be mildly reduced in *rad51-KN* and more so in *rad51-ED*. The apparent lack of statistical significance may reflect an insufficient sample size, if a statistical test was performed at all.

Response:

We conducted statistical analysis for Figure 2B using a t-test for equal means revealing a p-value of 0.017 between wildtype and *rad51-ED* showing the means are significantly different. However, the fold difference of 3.86 when comparing means and 3.0 when comparing median values is minor when juxtaposed with *rad52*, *rad51* and *rad54* mutants, which have about a 100-fold effect. A p-value of 0.084 between wildtype and *rad51-KN* showed no significant difference although there is a fold difference of 2.15 when comparing means and 1.9 when comparing median values. Finally, there is a significant difference between *rad51-ED* and *rad51-KN* with a p-value of 0.027. No other p-values showed any significance for the *ED* and *KN* mutants. We added this information to the text on page 7.

The authors have already shown that ATP hydrolysis stimulated by ssDNA is reduced in Rad51-ED and Rad51-KN (Fig. 5A), more severely in Rad51-ED. Rad51 focus formation in response to MMS is also defective in both mutants (Fig. S8). These results are consistent with their defects in ssDNA binding.

Response:

We agree with the reviewer, but the key point is that the defect with dsDNA is considerably stronger than with ssDNA. The defect in ATPase activity may or may not reflect an underlying defect in DNA binding. It is unclear whether Rad51 focus formation in response to MMS reports the ssDNA binding, dsDNA binding, or both.

In addition, Fig. 10A demonstrates that sister-chromatid joint formation is reduced in both *rad51-ED* and *rad51-KN*. Even in the *sgs1* background, the data suggest that inter-sister association is

compromised in these mutants, in agreement with my interpretation of Fig. 2B.

Response:

We agree with the reviewer and conclude in the manuscript that sister chromatid joint formation is reduced in both mutants. However, it is unclear whether ssDNA and/or dsDNA binding is required for this. Hence, the result does not directly relate to the interpretation of Figure 2 data.

Taken together, the evidence strongly supports the view that these two mutants are hypomorphic alleles with defects across a broader spectrum of HR functions than proposed by the authors, with ED more severely affected than KN. Given that Rad51 has long been known to function in post-replicative repair, it is not surprising that partial defects in Rad51 activity produce corresponding partial phenotypes.

Furthermore, multiple studies have already reported that dsDNA binding of Rad51 is critical for protecting stalled replication forks from nucleases (refs. 30, 31). This further diminishes the novelty of the current work.

Response:

We agree with the reviewers that these *rad51* mutants are hypomorphs, and we have explicitly taken back the claim of separation-of-function. The reviewer correctly points, and we discuss this in the manuscript, that refs. 30, 31 showed that binding to dsDNA is critical for fork protection by RAD51. The key novelty in our manuscript is that this affects pathway usage at stalled forks.

Minor points

(1) Line 219: there is no S2A (just S2).

Response:

Corrected, thanks for noting.

(2) Fig. S2: The figure appears to show raw CFU counts rather than viability, despite being titled as viability. Additionally, the elevated levels of *rad51*-KN under both induced and spontaneous conditions are confusing and should be more clearly explained in the main text.

Response:

The reviewer is correct. The KN data were erroneously swapped with the wt data. Thank you for pointing this out. We apologize for our oversight and corrected the figure as well as numerical data.

(3) Line 292: It would be useful to include ssDNA as well in the assay.

Response:

We have conducted the experiment suggested by the reviewer and added the data as Figure S7E-H. The results confirm that the defect in binding dsDNA is more pronounced than the defect on ssDNA.

(4) Line 424: Should this refer to Fig. S3B rather than S3A?

Response:

Corrected, thanks for noting.

(5) Line 425: The nature of the discrepancy mentioned is unclear and should be clarified.

Response:

We added additional clarification.

(6) Line 493: I was puzzled to find that the FigS11 does not show SCJ formation in the *rad51*

null mutant. Regarding the importance of including the rad51 null mutant in Fig10A. No matter how many times it has been done previously, including a negative control is essential because it sets the base line for the data acquisition and interpretation.

Response:

We eliminated the comparison to the rad51 null mutant.

(7) Fig. 10B: The phenotypes of strains expressing Rad51-KN-MN and Rad51-ED-MN should be compared directly with their untagged versions. The MN tag may alter the properties of these alleles.

Response:

We conducted this comparison and added the data to Figure S11A.

Reviewer #4 (Remarks to the Author):

I have reviewed the responses to each critique made by all three reviewers. I think that by changing the terminology from separation of function mutants was appropriate, they addressed the biochemical concerns adequately by performing their binding assays with new substrates and their D-loop assays at higher salt concentrations, and by removing their alpha fold model structural analysis the authors have more than adequately addressed all reviewers concerns- particularly Reviewer 2.

Response:

Thank you for your supportive comments.